# P300 promotes tumor recurrence by regulating radiation-induced conversion of glioma stem cells to vascular-like cells

Sree Deepthi Muthukrishnan[1], Riki Kawaguchi[1], Pooja Nair[1], Rachna Prasad[1], Yue Qin[1], Maverick Johnson[1], Qing Wang[1], Nathan VanderVeer-Harris[1], Amy Pham[1], Alvaro G. Alvarado[1], Michael C. Condro[1], Fuying Gao[1], Raymond Gau[1], Maria G. Castro[2], Pedro R. Lowenstein[2], Arjun Deb[3], Jason D. Hinman[4], Frank Pajonk[5], Terry C. Burns[6], Steven A. Goldman[7,8], Daniel H. Geschwind[1,4] & Harley I. Kornblum[1,9] ✉

Glioma stem cells (GSC) exhibit plasticity in response to environmental and therapeutic stress leading to tumor recurrence, but the underlying mechanisms remain largely unknown. Here, we employ single-cell and whole transcriptomic analyses to uncover that radiation induces a dynamic shift in functional states of glioma cells allowing for acquisition of vascular endothelial-like and pericyte-like cell phenotypes. These vascular-like cells provide trophic support to promote proliferation of tumor cells, and their selective depletion results in reduced tumor growth post-treatment in vivo. Mechanistically, the acquisition of vascular-like phenotype is driven by increased chromatin accessibility and H3K27 acetylation in specific vascular genes allowing for their increased expression post-treatment. Blocking P300 histone acetyltransferase activity reverses the epigenetic changes induced by radiation and inhibits the adaptive conversion of GSC into vascular-like cells and tumor growth. Our findings highlight a role for P300 in radiation-induced stress response, suggesting a therapeutic approach to prevent glioma recurrence.

Glioblastoma (GBM) is a universally recurrent and lethal primary brain tumor that is highly resistant to standard chemo- and radiation therapy[1,2]. Therapeutic failure and tumor relapse is partially attributed to a pre-existing resistant fraction of glioma stem cells (GSC) that exhibit a high degree of plasticity and molecular heterogeneity[3]. GSC are distinguished from differentiated tumor cells using putative cell surface markers such as CD133, CD44 and CD15. However, these markers do not strictly define GSC but only enrich for cells with stem-like features. GSC are primarily defined by their functional characteristics such as their ability to self-renew and initiate tumors, and the capacity for differentiation into multiple lineages[4]. GSC and non-GSC tumor cells can interconvert under certain metabolic and

[1]The UCLA Intellectual and Developmental Disabilities Research Center, David Geffen School of Medicine, UCLA, Los Angeles, CA, USA. [2]Department of Neurosurgery, and Department of Cell and Developmental Biology, University of Michigan Medical School, Ann Arbor, MI, USA. [3]Division of Cardiology, Department of Medicine, David Geffen School of Medicine, UCLA, Los Angeles, CA, USA. [4]Department of Neurology, David Geffen School of Medicine, UCLA, Los Angeles, CA, USA. [5]Department of Radiation Oncology, David Geffen School of Medicine, UCLA, Los Angeles, CA, USA. [6]Department of Neurological Surgery, Mayo Clinic, Rochester, MN, USA. [7]Center for Translational Neuromedicine, University of Rochester Medical Center, Rochester, NY, USA. [8]Center for Translational Neuromedicine, University of Coppenhagen School of Medicine, Coppenhagen, Denmark. [9]Department of Molecular and Medical Pharmacology, David Geffen School of Medicine, UCLA, Los Angeles, CA, USA. ✉e-mail: hkornblum@mednet.ucla.edu

environmental stress conditions. Temozolomide (TMZ) and radiation therapy-induced stress can lead to dedifferentiation of tumor cells to a GSC-like state, and these cells are termed induced-GSC (iGSC) and have been shown to promote tumor growth in xenograft models[5–9].

A growing body of evidence indicates that GSC can switch from a Proneural to Mesenchymal state in response to radiation therapy, and Mesenchymal-GSC contribute to tumor invasion and resistance[10,11]. GSC also transdifferentiate into vascular endothelial-like cells, albeit at a low frequency and contribute to vascularization and tumor growth[12–14]. Recent studies, however, indicate that GSC-derived endothelial-like cells increase in number in recurrent GBM[15,16] indicating that therapy may promote endothelial-like transition. Pericytes, which are mesenchymal in origin and form the other major component of the glioma vessels, can also arise from GSC, and their depletion disrupts tumor growth and maintenance of the blood tumor barrier[17,18]. It remains to be determined whether therapeutic stress can induce this pericyte-like phenotype in GSC and non-GSC tumor cells, and whether they contribute to tumor relapse.

While substantial progress has been made to understand the genetic and molecular events that drive tumor evolution and resistance, the precise molecular and epigenetic factors mediating adaptive phenotypic transitions of GSC and tumor cells in response to radiation- and chemotherapy-stress remains to be elucidated.

In this study, we utilized single-cell and whole-transcriptomic approaches to determine the contribution of radiation therapy-induced stress in reprogramming glioma cells to adopt diverse phenotypic states. We show that radiation stress promotes the phenotypic conversion of GSC to vascular endothelial- and pericyte-like-cells, which, in turn influence tumor growth and recurrence post-treatment. Radiation stress alters chromatin accessibility and H3K27 acetylation in specific vascular gene regions resulting in their increased expression, and this is mediated by the histone acetyltransferase (HAT) P300. Blocking P300 HAT activity reverses the epigenetic changes and vascular-like phenotype conversion and reduces tumor growth post-treatment, highlighting its potential as a therapeutic vulnerability for preventing GBM relapse.

## Results

### Single-cell transcriptomic sequencing reveals a dynamic shift in cellular states of glioma cells in response to radiation-stress

To determine the contribution of radiation-induced stress to phenotypic plasticity in GBM, we performed single-cell RNA-sequencing of a primary gliomasphere line exposed to a single dose of radiation at 8 Gy for 2- and 7-days. Integrated analysis of all the samples revealed 14 distinct clusters, a majority of which showed a dynamic shift in size (clusters 1, 2, 3, 5, 8, 10, 11), while a few increased (clusters 0, 4, 9) and some sharply reduced (clusters 6, 7, 12, 13) between 2- and 7-days post-radiation (Fig. 1a, b). We next performed an unbiased analysis using co-expressed gene networks and clustering by Louvain community detection to annotate clusters based on functional cell states. 29 gene-signature modules were identified that were differentially expressed between the 14 clusters, and clustering divided them into two major fractions that further separated into 4 sub-groups (Fig. 1c). Gene ontology (GO) analysis of the modules showed that subgroup S2 that contained clusters 6 and 7 (diminished post-radiation) were enriched for mitosis and DNA-repair processes. Subgroup S4, which comprised of clusters 0 and 4 (increased post-radiation) were highly enriched for vasculogenesis and mesenchymal stem cell differentiation modules. Subgroups S1 and S3 that contained clusters, which either increased[9], reduced[12–14] or dynamically changed[3,10,11] post-radiation showed enrichment for mitosis, stress-response, homeostasis and embryonic development-related processes, indicating the diverse functional states of these cells (Fig. 1c).

In order to define the functional lineage hierarchy of these cell states, we performed pseudotime trajectory analysis with

cluster 0 chosen as an arbitrary starting point. We found that on the one hand, subgroup S2 representing the stem cell-like state (in blue) branched into subgroups S1 (mitotic/proliferative-like state, in brown) and S3 (embryonic-like, in green), and on the other, gave rise to subgroup S4 (in red) enriched for vascular and mesenchymal-like processes (Fig. 1d). To further validate their identities, we examined known markers of proliferation (mitotic tumor cells), glioma stem cell-like (GSC), vascular (endothelial and mesodermal), and mesenchymal (pericyte/neural crest) cells in each of the clusters. We included neural crest and mesodermal markers because pericytes, which have been shown to arise from GSC in tumors, are derived from neural crest/mesenchymal stem cells (NC-MSC) in the brain, and endothelial cells are derived from mesodermal progenitor cells[17,19,20]. Consistent with the GO analysis results, cluster 0 from subgroup S4 highly expressed markers of GSC, NC-MSC and mature pericyte markers indicating that these may indeed represent mesenchymal/pericyte-like cell population. Cluster 4 belonging to the same subgroup S4 showed increased expression of some mesodermal, and many mature endothelial markers suggesting that these are endothelial-like cells (Fig. 1e). These clusters showed very little expression of proliferation markers, indicating that they are differentiated cells, rather than actively proliferative tumor cells. All clusters from subgroup S2 showed high expression of several GSC and proliferation markers confirming that these are stem-like cells. Clusters[9,13,14] in subgroup S1 showed moderate expression of proliferation markers but no GSC markers indicative of mitotic tumor cells. Clusters in subgroup S3 showed variable expression of GSC and embryonic markers suggesting that these might be a transient cell population as they change dynamically in size between 2- and 7-day post-radiation (Fig. 1e). These data indicate that a fraction of tumor cells and GSC that survive radiation-stress acquire the endothelial-like and mesenchymal, pericyte-like cell states.

To examine if the in vitro functional cell states are replicated in vivo, we performed scRNA-sequencing on tumor cells isolated from radiated and control xenografts. Integrated analysis of the samples identified 15 clusters in total, of which clusters 1,2-4,9,10, 12 and 13 increased in frequency, and clusters 0, 5, 6,7 and 11 diminished in radiated tumors. (Fig S1A, B). Similar to in vitro findings, Louvain community detection clustering revealed 2 major fractions and 4 sub-groups (Supplementary Fig. 1c). However, subgroup 3 was further divided into two: S3a and S3b as they showed enrichment for different sets of modules. Overall, clusters with greater frequency in radiated tumors were enriched in adhesion/junction assembly, cGMP signaling, angiogenesis, epithelial-mesenchymal transition, and migration-related processes, whereas clusters that diminished in size expressed modules enriched in embryonic eye development, ion homeostasis, migration, and stem cell differentiation (Supplementary Fig. 1c). Pseudotime trajectory and known marker analysis revealed that clusters[4,9] from subgroup S1 (blue) had highest expression of GSC markers representing the stem-like cells. Subgroup S3a (green) with clusters[10,13] showed increased expression of mitotic, as well as moderate expression of GSC, embryonic and vascular markers indicative of an intermediate cell state, and S3b with clusters[2,3,5] represented the mitotic tumor cells. Subgroup S2 (red) containing cluster 8 displayed expression of mesodermal and mature endothelial markers. Subgroup S4 expressed NC-MSC and pericyte markers, and both these clusters increased in size post-radiation (Supplementary Fig. 1d, e). The pericyte-like and endothelial-like clusters showed very low levels of proliferation markers indicating that they are differentiated cells. Together, these data indicate that radiation-refractory tumor cells acquire vascular endothelial-like and mesenchymal pericyte-like phenotypes in tumor xenografts, essentially replicating the in vitro findings.

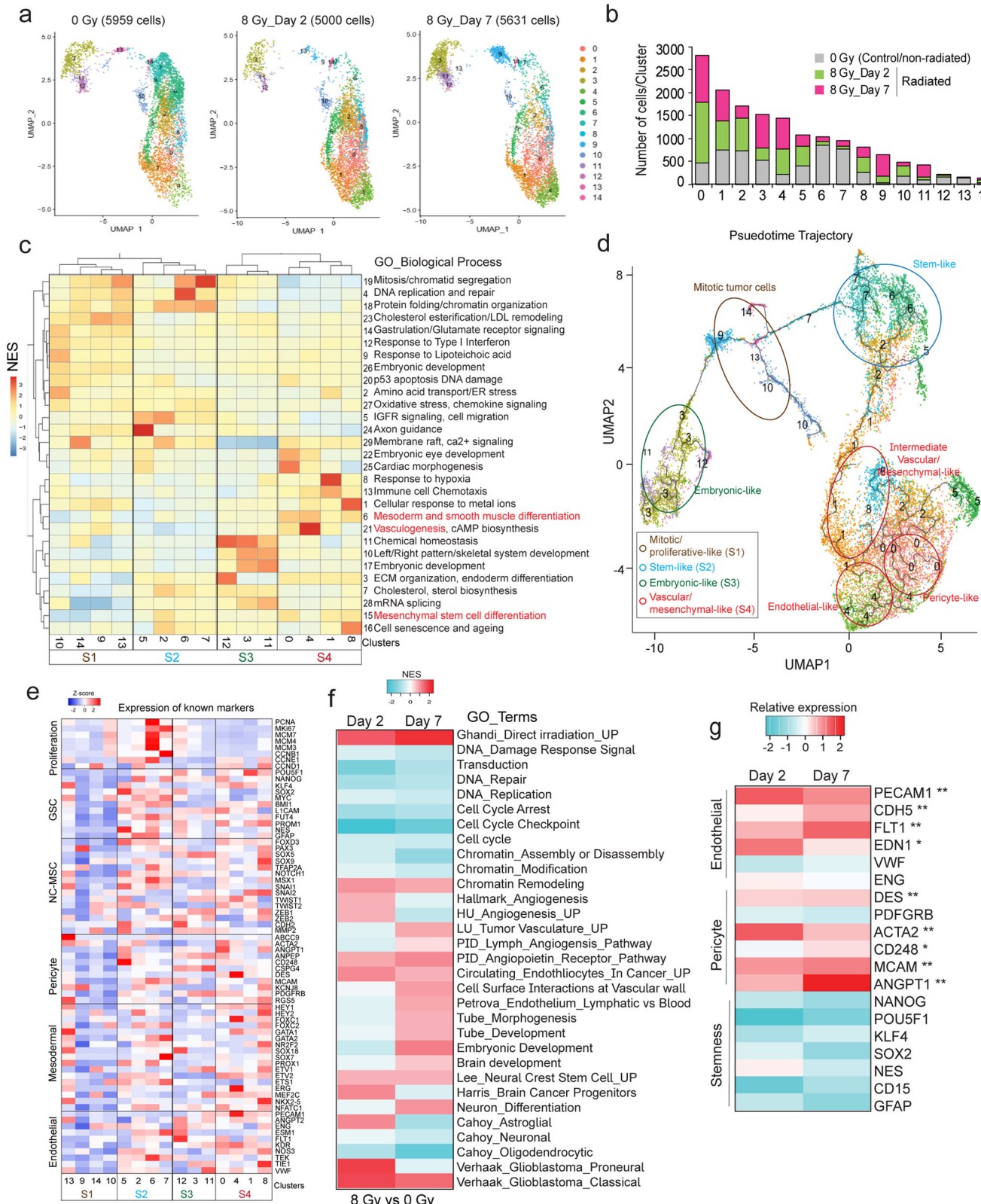

**Fig. 1 | Single-cell sequencing reveals the diverse functional states of glioma cells post-radiation. a** UMAP plot of cell clusters from control (0 Gy) and 2- and 7-days radiated (8 Gy) fractions. **b** Histogram shows the number of cells per cluster in each group. **c** Heatmap shows Louvain clustering of co-expressed gene network modules and EnrichR GO_terms associated with each cluster. Each subgroup is highlighted by a black box. **d** Pseudotime trajectory analysis, subgroups are highlighted in colored circles and their functional cell states indicated in the inset.

**e** Heatmap of average expression of markers of proliferation, glioma stem cells (GSC), Neural crest-mesenchymal stem cells (NC-MSC), pericytes, mesodermal and endothelial cells. **f** Heatmap of normalized enrichment scores (NES) of gene sets in glioma cells 2- and 7-days post-radiation. $N = 3$ biological replicates and only gene sets with $p < 0.05$ are shown. **g** Heatmap of relative expression of endothelial, pericyte and stemness genes in control and radiated cells. $N = 3$ independent experiments. *, ** indicates $p < 0.05$, $p < 0.005$, unpaired two-tailed t-test.

## Whole-transcriptomic sequencing confirms the enrichment of angiogenic and vascular markers in gliomaspheres post-radiation

Next, we sought to determine if radiation-stress-induced functional states observed in single-cell studies are discernible in patient-derived gliomasphere cultures by bulk RNA sequencing. Hierarchical clustering of normalized gene expression showed that radiated cells at day 2 and day 7 clustered separately from each other, and from control cells (Supplementary Fig. 1f, g). As expected, gene set enrichment analysis (GSEA) showed enrichment of GO: terms related to radiation, and reduction in gene sets associated with cell cycle and DNA repair in both 2- and 7-day radiated cells (Fig. 1f). In line with single-cell studies, we also found enrichment of gene sets related to tumor vasculature, angiogenesis, embryonic development, NC-MSC and mesenchymal transition, and downregulation of Proneural and neuronal gene sets in radiated gliomaspheres. Closer examination of the genes associated with these processes showed significant enrichment of angiogenesis genes (*p < 0.05, Benjamini-Hochberg adjusted*), and a small subset of mesenchymal genes from the Verhaak_Glioblastoma_Mesenchymal geneset in 7-day radiated gliomaspheres (Supplementary Fig. 1h, i). QRT-PCR confirmed the enrichment in vascular markers that included endothelial (PECAM1, CDH5, FLT1, EDN1) and pericyte (DES, ACTA2, ANGPT1, MCAM, CD248) genes in day 7 radiated gliomaspheres, which was further validated by immunostaining (Fig. 1g, Supplementary Fig. 1j). These markers were not significantly upregulated in day 2 gliomaspheres, corroborating the findings from bulk RNA sequencing. These data suggest that radiation stress primarily promotes vascular gene expression in gliomaspheres.

## Glioma cells exhibit increased endothelial and pericyte marker expression in response to radiation-stress

Standard radiation therapy for GBM patients involves a fractionated dose regimen of 2 Gy for 30 days[21]. However, our transcriptomic analyses were performed with a single high dose of radiation, 8 Gy. Therefore, we assessed whether fractionated radiation (2 Gy, x4) also induces vascular gene expression in gliomaspheres. QRT-PCR analysis showed significant increase in endothelial and pericyte markers, however, mesenchymal and GSC markers were not significantly upregulated by fractionated or single-dose radiation (Fig. 2a). In addition, both low (2 and 4 Gy) and high (8 and 10 Gy) dose were effective at inducing the expression of the majority of the vascular genes, but not GSC markers (Supplementary Fig. 2a).

To determine the generalizability of our findings, we used the single high dose 8 Gy to verify if radiation promotes vascular marker expression in multiple patient-derived gliomasphere lines that included the 3 major molecular subtypes: Proneural, Mesenchymal and Classical[22], and both primary and recurrent IDH1-wild type and mutant GBM. We found significant upregulation of vascular markers, albeit to a varying degree, in all lines examined irrespective of the subtype and mutational status. Interestingly, Proneural lines (HK_217, HK_408, HK_347) showed the highest expression for endothelial markers (PECAM1, CDH5), and the Mesenchymal lines (HK_372, HK_412, HK_308) exhibited the highest pericyte marker expression (DES, ACTA2). Although the primary GBM lines did not show an increase in GSC markers, some of the recurrent GBM lines showed an increase in GSC markers, including SOX2, NANOG and OCT4 post-radiation (Fig. 2b). These findings indicate that radiation stress can promote both stem-like and vascular-like marker expression, but the acquisition of stem-like state may be restricted to a subset of gliomas.

To quantify the extent of radiation stress-induced vascular phenotype conversion, we generated lentivirus constructs containing endothelial (VE-CADHERIN/CDH5) or pericyte (DESMIN) promoter to drive expression of a mCherry reporter. We found that maximal reporter activation from both promoters occurs 3- and 7-days post-radiation (Fig. 2c, d). Flow cytometric analysis using previously validated endothelial (CD31, CD144) and pericyte markers (CD146, CD248) also showed increased percentage of double-positive cells 3- and 7-days post-radiation (Supplementary Fig. 2b). Co-expression of both reporters showed very little overlapping expression of VE-CADHERIN-mCherry and DESMIN-GFP in both control and radiated gliomaspheres (Supplementary Fig. 2c). Immunostaining and qRT-PCR analysis of FACS sorted reporter-positive and negative fractions also indicated that endothelial and pericyte markers are expressed by distinct cells in gliomaspheres supporting the results from single-cell studies that they constitute different cell states (Supplementary Fig. 2d, e).

In order to distinguish between the possibilities that radiation: a) induces the vascular-like phenotypes, versus b) promotes the expansion of pre-existing vascular-like cells, we preemptively depleted the reporter-positive cells by FACS. Radiation of reporter-negative fractions activated both VE-CADHERIN and DESMIN reporters, but no activation was seen in non-radiated gliomaspheres (Supplementary Fig. 2f, g). We extended these findings by examining reporter activation in multiple patient-derived lines. Consistent with the gene expression data, while gliomaspheres from the Proneural (PN) and Classical (CL) subtypes showed similar expression levels for both reporters, the Mesenchymal (MES) subtype showed significantly higher pericyte reporter expression compared to endothelial reporter (Supplementary Fig. 2h), indicating that glioma cells from specific molecular subtypes may exhibit differential propensity for acquiring endothelial-like and pericyte-like states in response to radiation-stress.

## GSC exhibit higher propensity for vascular-like phenotype conversion than non-GSC in response to radiation-stress

Prior studies have demonstrated that CD133 + GSC transdifferentiate into endothelial cells and pericytes[12,17,23]. To determine if radiation promotes vascular-like conversion of GSC or non-GSC tumor cells, we also utilized CD133 (PROM1) as a GSC marker to separate the stem-like fraction from tumor cells[24]. Because this GSC marker is not informative in all tumors and cultures, we validated its significance in the cell line being used, HK_408 with limiting diluting assay and orthotopic transplantation of a small number (1000 cells) of CD133 + and CD133-cells into mice (Supplementary Fig. 2i-k). Radiation of either CD133 + or CD133- fractions showed that significantly increased endothelial and pericyte marker expression was restricted primarily to the CD133 + GSC fraction at 3 and 7-days post-radiation. GSC markers were also highly upregulated in the CD133 + GSC fraction relative to CD133- tumor cells (Fig. 2e). Immunostaining of endothelial (CD31/VE-CADHERIN) cells and pericyte (DESMIN/aSMA) markers, and reporter expression confirmed that GSC exhibit greater capacity for vascular-like phenotype conversion than non-GSC post-radiation (Fig. 2f, g). These findings indicate that at least for this gliomasphere line, that radiation-induced vascular-like phenotype is isolated to the stem cell fraction.

## Radiation promotes vascular endothelial- and pericyte-like phenotypes in orthotopic xenograft and murine GBM models

To determine if radiation promotes vascular-like phenotype conversion in vivo, we first generated orthotopic xenografts using the gliomasphere line HK_408 infected with a Firefly-Luciferase-GFP lentivirus. Tumor-bearing mice were exposed to a single dose of 8 Gy radiation. Immunostaining of vascular markers showed an increase in tumor cells co-expressing GFP with VE-CADHERIN or CD31 (endothelial) and DESMIN or αSMA (pericyte) predominantly in the tumor mass, and only 1-2 per section in the vessels, in radiated tumors compared to non-radiated tumors (Fig. 2h, Supplementary Fig. 2m). To quantitatively measure in vivo vascular-like conversion, we generated xenografts with tumor cells transduced with VE-CADHERIN (CDH5-mC) and DESMIN (DES-mC) reporters. Tumor-bearing brains from control and radiated groups were harvested 2 weeks after radiation, and analyzed by immunostaining and flow cytometry for the presence of GFP +

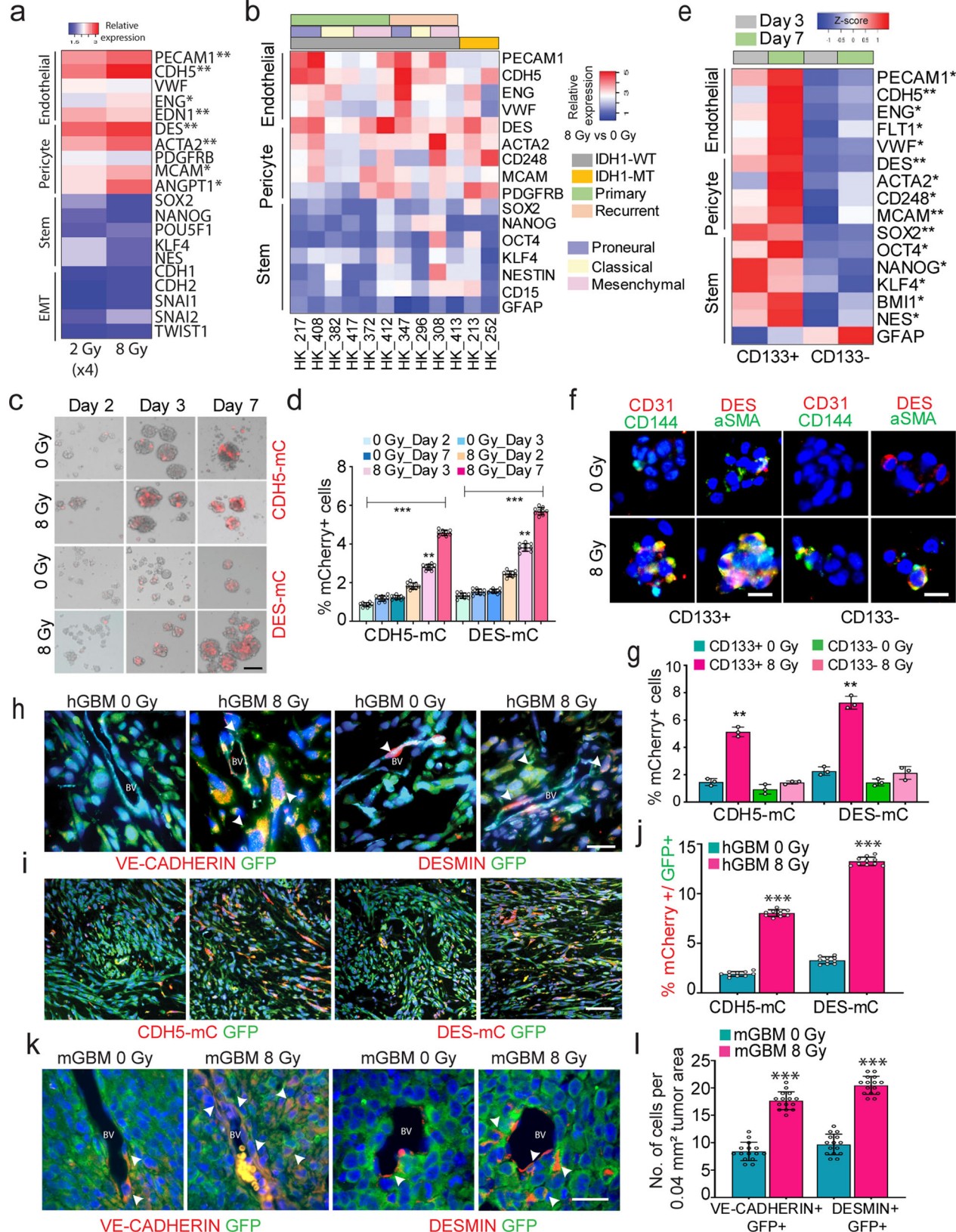

mCherry+ cells (Fig. 2i). Immunostaining for GFP (tumor) and mCherry (vascular-like) revealed an increase in co-labelled cells, and flow cytometric quantitation showed that this increase was significant for endothelial- (8% vs 2% control, *$p < 0.005$) and pericyte- (13% vs 3% control, **$p < 0.005$) like cells in radiated tumors (Fig. 2j). These findings, in line with in vitro studies, indicate that radiation-stress

promotes the acquisition of endothelial-like and pericyte-like phenotypes in glioma cells in vivo.

Next, we asked whether the presence of an intact immune system affects vascular-like conversion post-radiation in a syngeneic murine GBM (mGBM) model[25]. We first verified that mGBM cultures showed enrichment of vascular markers post-radiation (Supplementary

**Fig. 2 | Radiation-stress promotes endothelial and pericyte marker expression in glioma cells. a** Heatmap of relative expression of endothelial, pericyte, stemness and EMT genes in control (0 Gy), fractionated (2 Gy X 4) and single dose (8 Gy) radiated glioma cells. $N = 3$ biological replicates, * and ** indicates $p < 0.05$ and $p < 0.005$, unpaired two-tailed t-test. **b** Heatmap of relative expression of endothelial, pericyte and stemness markers in radiated vs control glioma cells from multiple patient-derived gliomasphere lines. **c, d** mCherry expression in control and 2,3, and 7-days radiated cells. Flow-cytometric quantitation of percentage of mCherry+ cells. Error bars represent mean ± SD, $N = 3$ biological replicates. ** indicates $p < 0.005$, one-way ANOVA. **e** Heatmap of relative expression of endothelial, pericyte and stemness genes in CD133 + (GSC) and CD133- (non-GSC) fractions 3- and 7-days postradiation. $N = 3$ biological replicates, * and ** $p < 0.05$, and $p < 0.005$, one-way ANOVA, post hoc t-test. **f** Immunostaining of endothelial (CD31, CD144/VE-CADHERIN) and pericyte (DES, aSMA) markers in control and radiated

CD133 + and CD133- fractions. Scale bars, 50 μm. **g** Quantitation of mCherry+ cells in control and radiated CD133 + and CD133- fractions. $N = 3$ biological replicates, error bars represent mean ± SD, ** $p < 0.005$, unpaired two-tailed t-test. **h** Immunostaining of VE-CADHERIN and DESMIN in GFP + tumor cells in control and radiated xenografts. Scale bars, 25 μm. White arrows point to GFP + marker+ cells in each image. **i, j** Immunostaining of mCherry and GFP in control and radiated tumor xenografts. Flow-cytometric quantitation of mCherry+ cells normalized to total number of GFP + tumor cells in each group. Error bars represent mean ± SD. $N = 10$ biological replicates, ** indicates $p < 0.005$ unpaired two-tailed t-test. Scale bars, 100 μm **k, l**. Immunostaining of VE-CADHERIN and DESMIN in GFP + tumor cells in murine GBM model. Arrows point to marker+ GFP + cells. Scale bars, 25 μm. Graphs show quantitation of number of GFP + marker+ cells in the tumor mass per section. $N = 3$ mice, * and *** indicates $p < 0.05$ and $p < 0.0005$ unpaired two-tailed t-test.

Fig. 2n). Examination of tumors showed that there was co-staining of GFP (tumor cells) with endothelial (VE-CADHERIN) and pericyte (DES) markers postradiation. Quantification of GFP + marker+ cells revealed that there was significant increase in vascular-like cells in radiated mice more predominantly (18–20 cells per tumor area) in the tumor mass, but also a small yet significant increase in (2 cells per vessel) both endothelial-like and pericyte-like markers in the vessels (Fig. 2k, l and Supplementary Fig. 2o, p). This indicates that radiation-induced vascular-like phenotype conversion is not inhibited by an intact immune microenvironment as would exist in human tumors.

## Radiation-induced glioma endothelial cells (iGEC) display phenotypic characteristics of normal vascular endothelial cells

Typically, the identity and behavior of normal vascular endothelial cells (VEC) is ascertained by their ability to uptake labelled low-density lipoproteins (LDL) and form tubular networks on GFR-Matrigel mimicking the blood vessels[26–28]. Earlier studies have shown that glioma-derived endothelial cells possess these characteristics[12–14]. We therefore asked if radiation-induced glioma endothelial cells, referred to henceforth as "iGEC" also exhibit characteristics of VEC. We used human umbilical vein endothelial cells (HUVEC) as a reference for normal VEC to assess these phenotypes. Similar to HUVEC, radiated glioma cells also showed tubular-network formation on Matrigel, and increased uptake of DiI-Ac-LDL compared to non-radiated cells (Fig. 3a-d, Supplementary Fig. 3a, b).

We next performed Matrigel plug in vivo angiogenesis assay using FACS sorted reporter-negative tumor cells (mCherry-) and or reporter-positive iGEC or nGEC (CDH5-mCherry +) to determine if they integrate into host vasculature or induce angiogenesis when embedded subcutaneously (Fig. 3e). Histological examination with hematoxylin & eosin (H&E) and Masson's Trichrome staining revealed vessel formation in both GBM and GEC-embedded plugs. Immunostaining revealed a few mCherry+ cells lining along the vessels labelled with Tomato Lectin only in iGEC and nGEC groups indicating that some vascular-like cells may integrate into host vasculature (Fig. 3f). Quantitation of Trichrome stained sections also showed that both iGEC and nGEC groups had significantly higher vascular density than their corresponding tumor fractions (Fig. 3g).

To validate that a) the angiogenic and vascular gene signatures are restricted to vascular-like cells, and b) they are molecularly distinct from the rest of the tumor cells, we performed RNA-sequencing on sorted reporter + (iGEC/GPC and nGEC/GPC, mCherry +) and reporter- (mCherry-) tumor cells from control and radiated gliomaspheres. Multidimensional scaling (MDS) of gene expression showed that tumor cells and vascular-like cells clustered separately (Fig. 3h). DEA also revealed significant number of up and down-regulated genes between tumor cells vs. vascular-like cells (Supplementary Fig. 3c). Examination of canonical endothelial (CD31 and VE-CADHERIN) and pericyte (DES and ACTA2) markers confirmed the enrichment of these transcripts in GEC and GPC fractions, respectively (Supplementary

Fig. 3d). GSEA also showed increase in vasculature and angiogenesis-related gene sets in both non-induced and induced GEC and GPC fractions compared to their respective tumor fractions (Fig. 3i). GPC fractions were also enriched for smooth muscle- and mesenchymal-gene signatures consistent with their mesenchymal identity (Fig. 3i). We also observed downregulation of stem cell, neuronal, glia-related gene sets in the vascular-like fractions relative to tumor cells, further confirming that these differentiated vascular-like cells are molecularly distinct from their precursor tumor cells, and exhibit some vascular-like characteristics.

## iGEC and iGPC provide trophic support to promote tumor growth post-radiation

Given that only a small number of iGEC and iGPC contribute to vessel formation, and majority are located in the tumor mass, we asked whether they provide a trophic niche to promote tumor growth following treatment. To test this hypothesis, we first collected conditioned media (CM) from FACS sorted and cultured non-induced (nGEC/nGPC) and induced (iGEC/iGPC) vascular-like cells (Fig. 4a). Addition of CM from iGEC/iGPC to radiated tumor cells significantly promoted their proliferation, whereas it did not show significant growth-promoting effect on nonradiated tumor cells. CM from unsorted GBM cells also did not promote growth of radiated tumor cells (Fig. 4b). We also performed this assay on control and radiated tumor cells isolated from tumor xenografts, and obtained similar results, confirming that the vascular-like cells provide trophic support to radiated tumor cells (Supplementary Fig. 4a).

Next, we explored the transcriptomic data of sorted non-induced and induced GEC and GPC fractions to determine candidate trophic factors expressed by these transdifferentiated cells that can promote the proliferation of radiated tumor cells. We found several mitogenic growth factors, cytokines, and chemokines enriched in both fractions, as well some that were differentially enriched between GEC and GPC (Fig. 4c). We then directly tested the trophic potential of a few of these GEC/GPC-factors in promoting the proliferation of tumor cells in vitro. Of the commonly enriched factors, FGF7, but not IL1B, showed significant effect on the proliferation of radiated tumor cells (Fig. 4d). On the other hand, the factors uniquely enriched in GEC (IL23A and WNT16B) and GPC (IL33 and SEMA7A) significantly promoted proliferation of radiated (8 Gy mC-) tumor cells only (Fig. 4d). These results indicate that iGEC and iGPC secrete trophic factors that can promote the growth of tumor cells post-radiation stress.

To validate the trophic function of iGEC and iGPC in vivo, we employed two complementary strategies. First, we performed co-transplantation of control and radiated GFP-Luciferase expressing (mCherry negative) tumor cells with their respective mCherry-expressing (GFP-Luciferase negative) GEC and GPC fractions in a 1:1 ratio and assessed tumor growth at 2 and 4 weeks after transplantation (Fig. 4e). Co-transplanted iGEC and iGPC significantly enhanced the growth of radiated tumor cells (mC-) compared to tumor cells

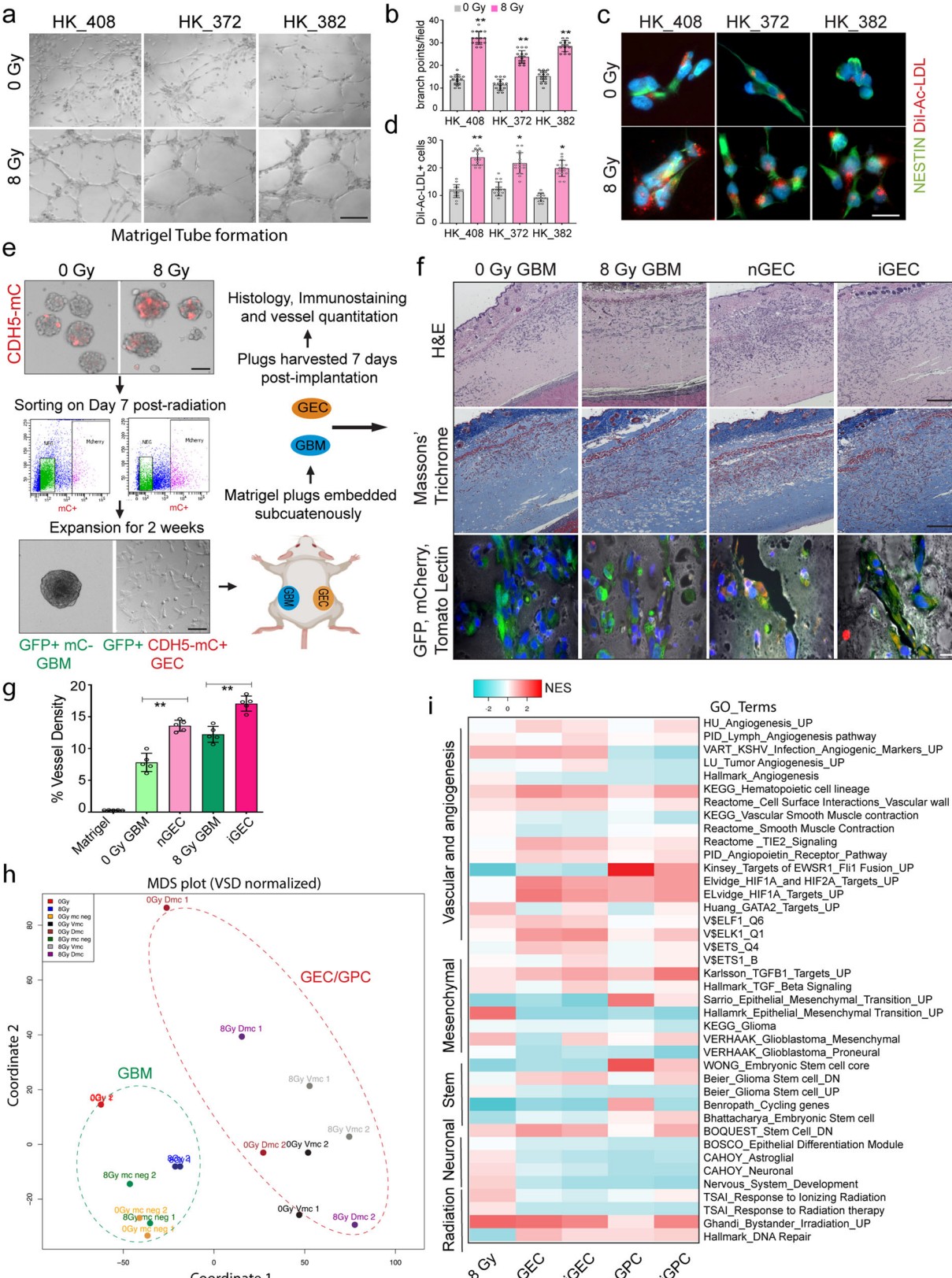

**Fig. 3 | iGEC display typical characteristics of normal vascular endothelial cells.** **a**–**d** Tubular network formation on matrigel and immunostaining of NESTIN and Di-Ac-LDL in control and radiated glioma cells. Scale bars, 100 μm and 50 μm. Quantitation of branch points and number of Di-Ac-LDL + cells per field in each group. Error bars represent mean ± SD, $N = 3$ biological replicates, * and ** indicates $p < 0.05$ and $p < 0.005$, unpaired two-tailed t-test. **e** Schematic outlines the matrigel plug in vivo angiogenesis assay. **f, g** H&E and Masson's trichrome staining. Immunostaining of mCherry (GEC, red), GFP (tumor cells, green) and Tomato Lectin (vessels, grey). Quantitation of vessel density. Error bars represent mean ± SD. $N = 5$ mice. *, ** indicates $p < 0.05$ and $p < 0.005$, one-way ANOVA. Scale bars, 500 μm and 20 μm. **h** MDS plot of clustering of tumor and transdifferentiated cells. **i** Heatmap of NES of GO_terms ($p < 0.05$) in each group.

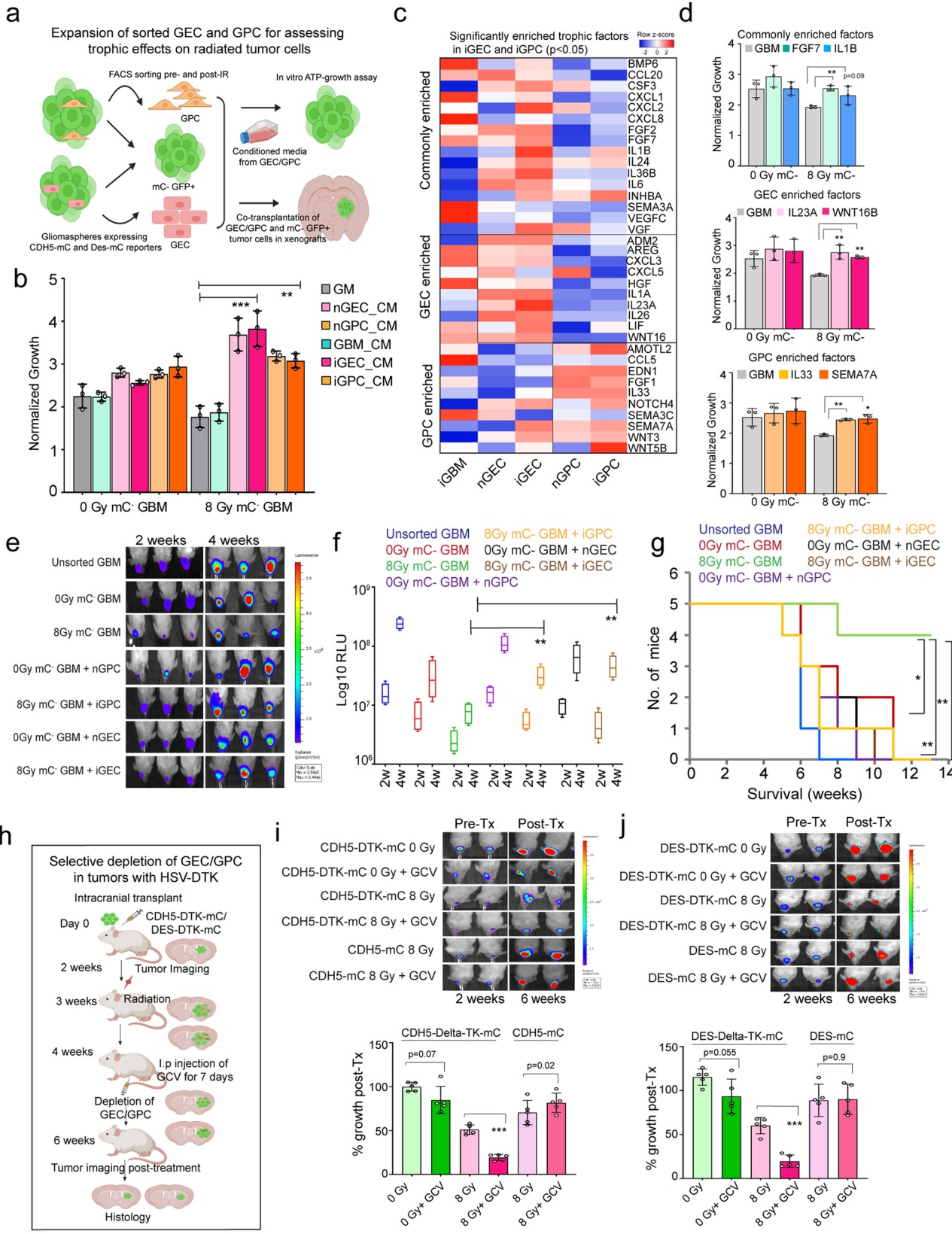

transplanted alone, corroborating the in vitro findings (Fig. 4f, Supplementary Fig. 4b). Survival analysis also showed that the co-transplanted mice exhibited signs of morbidity significantly earlier than mice transplanted with tumor cells alone (Fig. 4g).

Next, we performed selective depletion of iGEC/iGPC in vivo using the HSV-DTK cell ablation strategy by driving expression of HSV-DTK

under the control of either VE-CADHERIN (CDH5-HSVT-DTK-mCherry) or DESMIN (DES-HSV-DTK-mCherry) (Fig. 4h). After verifying tumor formation, mice were radiated with a single dose of 8 Gy and injected with Ganciclovir (GCV) every day for a week to allow depletion of reporter+ cells expressing HSV-DTK. Quantification of tumor growth pre- and postradiation and GCV administration revealed that depletion

**Fig. 4 | iGEC and iGPC provide trophic support to promote tumor growth post-radiation. a** Schematic of expansion of iGEC and iGPC for in vitro conditioned media (CM) and in vivo co-transplantation tumor growth experiments. **b** Proliferation of control and radiated tumor cells. Error bars represent mean ± SD, $N = 3$ biological replicates, ** and *** indicates $p < 0.001$ and $p < 0.0005$, one-way ANOVA and post hoc t-test. **c** LogFc expression (FDR adjusted, $p < 0.05$) of trophic factors expressed by radiated (iGEC/iGPC) and nonradiated (nGEC/nGPC) and tumor cells. **d** Proliferation of sorted, nonradiated (0 Gy mC-) and radiated (8 Gy mC-) tumor cells 3-days post-treatment with recombinant factors. $N = 3$ biological replicates, * and ** indicates $p < 0.05$ and $p < 0.005$, one-way ANOVA. **e, f** Images of

mice showing tumor growth at 2 and 4 weeks. Box plots show quantitation of tumor growth by luminescence in each group. Box plots display median, 25th and 75th percentile, and whiskers extend min and max values. $N = 5$ mice, * indicates $p < 0.05$, one-way ANOVA and post hoc t-test. **g** Kaplan-Meier survival curve of mice transplanted with tumor cells and GEC/GPC. $N = 5$ mice, and *, ** indicates $p < 0.05$, $p < 0.005$, Log-rank test. **h** Schematic outlines the depletion of GEC/GPC using HSV-DTK method in tumor xenografts **i, j** Images of mice showing tumor growth at 2 and 4 weeks. Box plots show the quantitation of tumor growth. $N = 5$ mice, *** indicates $p < 0.005$, one-way ANOVA and post-hoc t-test.

of either iGEC or iGPC markedly reduced tumor growth after radiation (Fig. 4i, j). On the other hand, the depletion of nGEC and nGPC did not show a significant effect on non-radiated tumors. We also did not see growth inhibition of tumors transduced with reporter constructs lacking HSV-DTK when administered with GCV indicating that only vascular-like cells expressing HSV-DTK were selectively depleted resulting in growth inhibition (Supplementary Fig. 4c, d). We also utilized another cell ablation strategy by expressing Diphtheria Toxin Receptor (DTR) to selectively deplete vascular-like cells. Since human cells are known to express the DT receptor, HB-EGF, we first ensured that the protein is not expressed by our cell line, and also verified that Diphtheria toxin (DT) treatment selectively ablated the DTR-expressing cells (Supplementary Fig. 4e, f). Similar to HSV-DTK depletion, mice-bearing tumors were radiated and administered with DT in 3 doses over the course of one week to ablate GEC/GPC (Supplementary Fig. 4g), resulting in significant inhibition in tumor growth (Supplementary Fig. 4h-j). Collectively, the in vitro and in vivo findings strongly suggest that vascular-like cells provide trophic support to tumor cells and promote recurrence postradiation treatment.

## Glioma cells exhibit increased chromatin accessibility and H3K27Ac in specific vascular gene regions post-radiation

Transition between cell states requires epigenetic and transcriptional reprogramming driven by alterations in chromatin structure and accessibility[29,30]. We therefore asked if radiation stress-induced vascular-like phenotype conversion involves changes in chromatin accessibility, especially in vascular gene regions. To test this hypothesis, we performed ATAC-sequencing on control and radiated gliomaspheres 2-days post-radiation. We chose the 2-day time point to determine changes that occur in the surviving fraction of tumor cells immediately after radiation-induced damage and prior to significant induction of vascular marker expression. Although there was a trend towards reduced chromatin accessibility in radiated cells, this difference was not significant ($N = 8$ samples, $p = 0.3$, unpaired two-tailed t-test (Fig. 5a). This was also reflected in percentage of peak distribution across various genomic regions, which did not show any overt differences (Fig. 5b).

Next, we examined if vascular genes that showed increased expression post-radiation displayed altered chromatin accessibility. Differential analysis revealed specific sites in endothelial (CDH5) and pericyte (ANGPT1) genes to be more open (indicated by increase in peak size) in radiated cells (Fig. 5c). This difference in peak size was not reflected in all genes. For example, the two housekeeping genes, RPL30 and GAPDH did not display altered accessibility between control and radiated cells (Fig. 5c). We also found that there are known H3K27Ac and transcription factor binding sites (TFBS) in these genomic locations in CDH5 and ANGPT1 genes in the UCSC genome browser, indicating that these regions could be essential for their transcriptional activation (Supplementary Fig. 5a). GO analysis of all the differentially open peaks revealed significant enrichment of terms associated with vascular development, endothelial differentiation, mesenchymal and stem cell development (Fig. 5d, Supplementary Fig. 5b). HOMER motif analysis also revealed significant enrichment of endothelial specification and mesenchymal transition-associated

transcription factor motifs in radiated cells (Supplementary Fig. 5c). These results suggest that radiation increases chromatin accessibility specifically in certain genomic regions of the vascular genes.

Histone acetylation is a key determinant of chromatin accessibility[31,32]. We wondered whether radiation altered histone acetylation levels leading to changes in chromatin accessibility in vascular genes. To address this, we performed immunoblotting for H3K27Ac and total H3 levels at different time points after radiation. We saw that the total histone 3 (TH3) levels were reduced at 6 hrs, but significantly increased around 24-48 hrs post-radiation. We also observed an increase in H3K27Ac levels at the same points (Fig. 5e, f). Immunostaining for total H3 and H3K27Ac in radiated and control tumor xenografts revealed a significant reduction in total-H3 positive tumor cells, and on the other hand, a significant increase in AcH3 positive cells post-radiation (Fig. 5g, h). We also examined radiated and control mGBM tumors and found significant increase in AcH3-positive tumor cells post-treatment (Supplementary Fig. 5d). Finally, we assessed whether the vascular gene regions identified in ATAC-sequencing showed H3K27Ac in radiated gliomaspheres. ChIP-qPCR analysis showed a significant increase in the enrichment of genomic regions associated with CDH5 and ANGPT1, but no difference in the housekeeping gene RPL30 in radiated tumor cells immunoprecipitated with anti-H3K27Ac antibody (Fig. 5i). These findings indicate that radiation-resistant tumor cells display increased accessibility and H3K27Ac levels in regions associated with vascular genes. Next, we examined whether CDH5 and ANGPT1 played a functional role in radiation-induced vascular conversion. Knockdown of either CDH5 or ANGPT1 with lentiviral expression of shRNAs reduced the expression of endothelial and pericyte markers induced by radiation, respectively. However, CDH5 knockdown did not alter pericyte markers, and ANGPT1 knockdown did not significantly alter endothelial marker expression, suggesting that these genes act as lineage markers of distinct vascular-cell states, and promote radiation-induced endothelial- and pericyte-like transdifferentiation, respectively (Fig. 5j, k).

## Blocking P300 histone acetyltransferase activity inhibits radiation-induced vascular-like phenotype conversion of glioma cells

Histone acetyltransferases (HAT) catalyze the transfer of acetyl groups to core histones regulating chromatin structure and gene transcription[33]. Since radiation alters H3K27ac in vascular genes, we sought to determine whether blocking histone acetylation prior to radiation inhibits vascular-like phenotype conversion. First, examination of our RNA-sequencing data revealed that of the 14 known HATs, the EP300 (KAT3B) and KAT2B transcripts showed uniformly enhanced expression in radiated tumor cells at both 2 and 7-days post-treatment. (Fig. 6a). EP300 was also enriched in GEC and GPC (Supplementary Fig. 6a). Hence, we decided to target the HAT activity of P300 using C646, a selective small molecule inhibitor[34]. We validated the inhibitor by examining its effect on H3K27ac by immunoblotting (Fig. 6b). QRT-PCR analysis of endothelial and pericyte markers showed significant downregulation in combined C646- and radiation-treated glioma cells compared to radiation alone (Fig. 6c). This inhibitory effect was seen in two other hGBM lines as well as in the mGBM

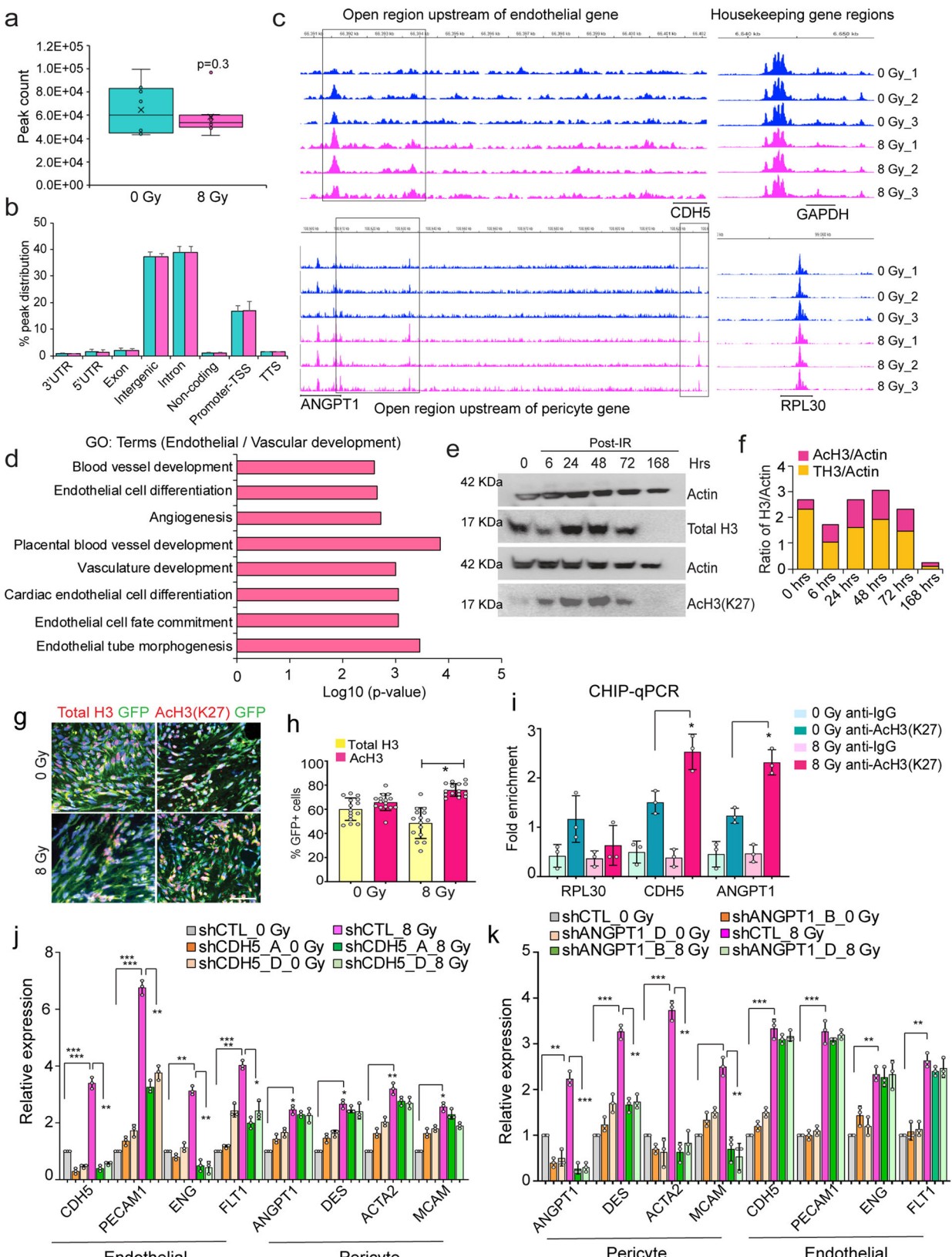

model (Supplementary Fig. 6b-d). We also found significant inhibition of radiation-induced vascular marker expression in CD133 + GSC fraction with C646 treatment (Fig. 6d, Supplementary Fig. 6e). Furthermore, we utilized the endothelial (CDH5-mCherry) and pericyte (DES-mCherry) reporters and determined that pre-treatment with C646 significantly reduced the expression of the reporter post-

radiation, as well as in sorted CD133 + GSC fraction (Fig. 6e, f and Supplementary Fig. 6f, g).

To determine whether P300 HAT activity is required for radiation-induced chromatin accessibility and H3K27ac of vascular gene regions, we performed ATAC-sequencing and ChIP-qPCR on control and radiated gliomaspheres treated with C646. Surprisingly,

**Fig. 5 | Radiation alters chromatin accessibility and H3K27ac in vascular gene regions. a, b** Total peak count and peak distribution across various genomic regions in 2-day control and radiated gliomaspheres. $N = 8$ biological replicates, and $p = 0.3$ derived from Welch's unpaired t-test. Box plots indicate median, 25th and 75th percentile, and whiskers extend from min to max value. **c** Differentially open regions (highlighted in black boxes) upstream of vascular genes (CDH5, endothelial and ANGPT1, pericyte), and housekeeping genes (RPL30 and GAPDH) in nonradiated and radiated cells. **d** Enrichment of vascular and blood vessel development related GO_terms in radiated gliomaspheres. **e, f** Immunoblot of total and Ac-histone 3 (AcH3, Lysine K27) in control and radiated cells. Quantitation of protein levels normalized to Actin is shown in the graph. **g, h** Immunostaining of Total H3 (red), AcH3 (red) and GFP (green) in control and radiated xenografts. Quantitation of GFP + marker+ positive cells per tumor section. Error bars represent mean ± SD. $N = 3$ mice. * $p < 0.05$, unpaired two-tailed t-test. Scale bars, 100um. **i** Fold enrichment of gene regions of CDH5 and ANGPT1 immunoprecipitated with anti-H327Ac and control anti-IgG antibody in control and radiated cells. RPL30 was used as a positive control. $N = 3$ biological replicates, * indicates $p < 0.05$ derived from one-way ANOVA. **j, k** Relative expression of endothelial and pericyte markers in non-radiated (0 Gy) and radiated (8 Gy), control (shCTL) and CDH5 (shCDH5_A and D) knockdown and ANGPT1 (shANGPT1_B and D) knockdown cells. $N = 3$ biological replicates, *, ** and *** indicates $p < 0.05$, $p < 0.005$, and $p < 0.0005$, one-way ANOVA and post-hoc t-test.

we found a significant increase in peak counts, and peak distribution specifically in intergenic regions with combined radiation + C646 treated glioma cells, whereas there were no significant differences amongst control, C646 or radiation treatment alone. PCA showed clear separation of radiated cells from radiation + C646 treated cells, and from control gliomaspheres (Supplementary Fig. h-j). We also found significant differences in the number of both up- and down-regulated genes associated with the peaks between radiation + C646 treated cells vs radiation alone (Supplementary Fig. 6k), indicating that pre-treatment with P300 HATi can profoundly affect radiation-induced changes in chromatin.

We next examined whether C646 treatment specifically reversed the effects of radiation-induced increase in chromatin accessibility in vascular genes. Indeed, the genomic regions in CDH5 and ANGPT1 that showed increased accessibility in radiated cells were reduced with C646 pretreatment (Fig. 6g). As expected, GO analysis of differentially open regions showed significant enrichment of terms associated with vascular development, and mesenchymal transition in radiated cells. However, the combined C646 + radiation treatment did not show enrichment of vascular development-related terms, but instead showed enrichment of terms related to mitosis, neuron development and chromatin organization and adhesion assembly (Supplementary Fig. 6l). To demonstrate that P300-mediated H3K27ac is required for radiation-induced changes in chromatin of vascular genes, we performed ChIP-qPCR with anti-H3K27ac antibody on radiated and C646 + radiation treated gliomaspheres. Pre-treatment with C646 significantly reduced the enrichment of genomic sites in CDH5 and ANGPT1, but not the housekeeping gene RPL30 in radiated cells (Fig. 6h, Supplementary Fig. 6m), thus confirming that P300 HAT activity is essential for radiation stress-induced epigenetic rewiring in glioma cells.

To determine if alterations in chromatin states in vascular genes is reflected in their transcriptional activation, we performed RNA-sequencing on glioma cells pre-treated with combined C646 and radiation or either treatment alone. In line with ATAC-sequencing results, treatment with C646 significantly changed gene expression in control and radiated cells, and PCA showed separation of radiated and C646 + radiation-treated cells (Supplementary Fig. 6n, o). Notably, pre-treatment with C646 reduced the expression of several vascular genes induced by radiation (Fig. 6i). We also found significant downregulation of growth factors enriched in iGEC and iGPC in C646 + radiation-treated cells, which was further validated by qRT-PCR (Fig. 6j, Supplementary Fig. 6p). GSEA demonstrated that gene sets related to vascular development, angiogenesis, and mesenchymal transition were diminished, and those associated with DNA repair were enriched (Fig. 6K, Supplementary Fig. 6q) with treatment. Taken together, these findings establish that P300 HAT activity is essential for radiation-induced epigenetic rewiring and vascular-like phenotype conversion of glioma cells.

**EP300-deficient glioma cells show reduced vascular-like phenotype conversion and tumor growth post-radiation in vivo**
To verify that P300 is essential for radiation-induced vascular-like conversion in vivo, we utilized lentiviral-shRNAs to knockdown EP300

in gliomaspheres following validation of knockdown efficiency and specificity pre and postradiation (Supplementary Fig. 7a, b) and its ability to diminish immunostaining of the P300 protein (Fig S7C). EP300-deficient cells (shEP300) showed reduced AcH3 compared to control (ShCTL/scrambled) cells (Fig. 7a, b). Consistent with the C646 inhibitor studies, EP300-deficient cells showed reduced endothelial and pericyte gene expression and reporter activation post-radiation (Fig. 7c-e, Supplementary Fig. 7d). Growth factors enriched in iGEC and iGPC also showed reduced expression in radiated, EP300-deficient cells (Fig. 7f, Supplementary Fig. 7e). Finally, we generated tumor xenografts with EP300-deficient and CTL cells and subjected them to radiation. EP300-deficient tumors showed significant reduction in growth post-treatment and increased animal survival (Fig. 7g, h). Immunostaining for total and AcH3 showed fewer AcH3-positive tumor cells with EP300-knockdown, and quantitation revealed that this reduction was significant in radiated, EP300-deficient tumors (Supplementary Fig. 7f, g). Immunostaining and quantitation of endothelial (VE-CADHERIN/CD31) and pericyte (DESMIN/aSMA) marker expression in tumor cells (GFP) showed a significant reduction in radiated, EP300-deficient tumors compared to radiated and untreated EP300-deficient or control tumors (Fig. 7i, j and Supplementary Fig. 7h, i). In summary, these findings indicate that P300 mediates the radiation-induced vascular-like phenotype acquisition of glioma cells (Fig. 8).

## Discussion
In this study, we show at the single-cell level that radiation stress significantly alters the functional states of glioma cells. Primarily, radiation induces the phenotypic transition of GSC into vascular endothelial-like and pericyte-like cells, which in turn provide trophic support to the radiated tumor cells and promote recurrence. Radiation stress induces the vascular-like phenotype conversion by altering the accessibility of the chromatin in specific vascular gene regions leading to their increasing expression. This phenotypic conversion can be blocked by inhibiting the HAT activity of P300, which highlights a key role for HAT in regulating treatment-induced plasticity and glioma recurrence.

Prior studies have indicated that glioma-derived endothelial cells can rarely incorporate into the vasculature, although their role in carrying blood has not been proven[12,14]. Glioma-derived pericytes are more common and their functional roles in glioma progression have been more clearly defined[17,18]. Our findings indicate that while radiation markedly enhances the frequency of GEC and GPC, these cells do not appear to incorporate into the vasculature to any great extent to play a key role in carrying blood supply. Rather, our data suggest that these cells provide the remaining tumor cells with trophic support, allowing them to survive the severe stress of radiation. The concept of a "vascular niche" that provides trophic support is a common finding in stem cell biology. In the mammalian central nervous system, endothelial cells provide a niche that supports the survival and self-renewal capacity of neural stem cells[35]. Studies in rodent models indicate that the vascular niche allows tumor cells to survive radiation-induced stress[36,37]. Our results indicate that radiation may cause gliomas to, in

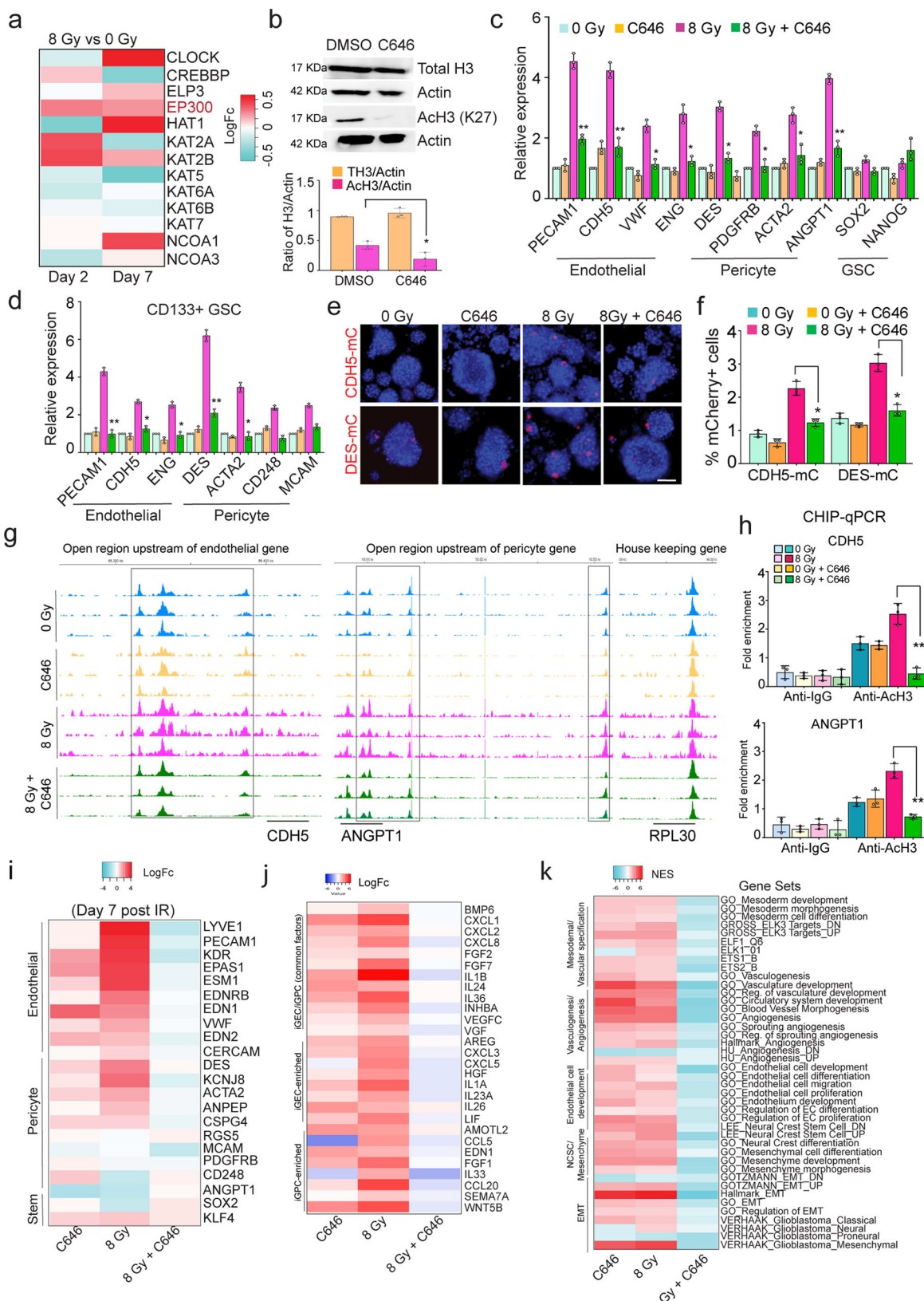

fact, create their own supportive niche by secreting trophic factors. While we identified some of these candidate factors and performed preliminary experiments to demonstrate their potential role in promoting proliferation of radiated tumor cells in vitro, further experiments will be needed to fully elucidate the underlying mechanisms governing their trophic function. The factors that mediate the trophic actions of radiation induced-vascular-like cells would likely be targets for potential therapeutic intervention to prevent GBM relapse.

Several mechanisms of neovascularization have been identified in GBM in addition to angiogenesis, and include vessel co-option, vascular mimicry and endothelial- and pericyte-transdifferentiation. These alternative neovascularization mechanisms are induced in

**Fig. 6 | Blocking P300 HAT activity inhibits radiation-induced epigenetic changes and vascular-like conversion in glioma cells. a** Heatmap of logFc expression of HATs in cultured control and radiated gliomaspheres. **b** Immunoblots of Total and AcH3 in DMSO and C646 treated gliomaspheres. Quantitation of protein level is shown in the graph. $N = 3$ biological replicates, and * indicates $p < 0.05$, unpaired two-tailed t-test. **c, d** Relative expression of endothelial, pericyte and GSC markers in control and radiated gliomaspheres and sorted CD133 + GSC fractions treated with C646. Error bars represent mean ± SD, $N = 3$ biological replicates. * and ** indicates $p < 0.05$ and $p < 0.005$, one-way ANOVA and post hoc t-test. **e, f** mCherry expression in control and radiated cells treated with C646. Flow-cytometric quantitation of mCherry+ cells. $N = 3$ biological replicates,

* indicates $p < 0.05$, one-way ANOVA. **g** Differentially open peak regions (highlighted in black boxes) in vascular genes CDH5, ANGPT1 in control and radiated cells alone or treated with C646. **h** Fold enrichment of CDH5 and ANGPT1 genomic regions immunoprecipitated with anti-H3K27Ac and control anti-IgG antibodies, normalized to input control $N = 3$ biological replicates, ** indicates $p < 0.005$, one-way ANOVA. **i** LogFC expression ($p < 0.05$) of endothelial, pericyte and stemness genes in control and radiated glioma cells treated with and without C646. **j** LogFC expression ($p < 0.05$) of iGEC and iGPC-enriched growth factors control and radiated glioma cells treated with and without C646. **k** Heatmap shows significant NES ($p < 0.05$) of gene sets in control and radiated glioma cells treated with and without C646.

response to anti-angiogenic therapy or by standard treatments of radiation and TMZ and contribute to therapeutic resistance[38,39]. Recent studies have also reported numerous signaling factors involved in promoting vascular transdifferentiation of GSC such as HIF1A, NOTCH1, ETV2, WNT5A, TIE2 signaling for endothelial-like conversion, and TGF-β signaling for pericyte-like transdifferentiation[12–15,17,40,41]. However, our transcriptomic analyses did not show enrichment of these factors in either endothelial-like or pericyte-like clusters or tumor cells post-radiation. This indicated that radiation-induced vascular-like conversion in GSC may occur via a different molecular mechanism. Radiation has been shown to alter histone gene expression and induce methylation changes in both cell lines and in animal studies[42,43]. Because cell state transitions and phenotype switching requires alterations in the epigenome, we posited that epigenetic rewiring during DNA repair following radiation-induced damage may promote phenotype plasticity in glioma cells. Using ATAC sequencing we found that while there was no consistent change in global accessibility, chromatin was markedly altered in vascular gene regions in radiated cells in all the samples. GO analysis of differentially open regions also revealed enrichment of terms associated with vascular development, endothelial differentiation and mesenchymal transition, supporting our hypothesis that chromatin rewiring in specific vascular genes by radiation-stress drives the vascular-like conversion of glioma cells.

Lysine acetylation of histones is a key post-translational modification that regulates chromatin accessibility and gene expression[30]. We found increased levels of H3K27ac in vascular gene regions in radiated cells, which indicated that blocking histone acetylation may inhibit the phenotype conversion induced in response to radiation stress. Histone acetyltransferases (HAT) catalyze the transfer of acetyl groups onto core histones altering the chromatin structure[33]. Of all the HAT genes, P300/KAT3B was enriched in radiated tumor cells and vascular-like cells. P300 plays a critical role in the DNA damage response by facilitating repair at sites of double-strand breaks (DSB) through acetylation of histones and chromatin decompaction[44–47]. Selective inhibition of P300 HAT activity using C646 small molecule inhibitor has been shown to inhibit cell growth and sensitize cells to DNA damaging agents in other cancers, including melanoma[48] NSLC[49], colorectal[50], prostate[51,52] and neuroblastoma[53]. Here, we found that blocking P300 HAT activity resulted in significant reduction of chromatin accessibility and H3K27ac in vascular gene regions in response to radiation. However, these effects were highly site-specific to vascular genes induced by radiation, as global chromatin accessibility counterintuitively increased with the combined C646 and radiation treatment. RNA-sequencing results also mirrored the ATAC-sequencing data as C646 treatment reduced vascular gene expression and gene sets associated with angiogenesis induced by radiation, but conversely increased metabolic and cellular biosynthesis-related processes that decreased with radiation. These results suggest that P300 HAT activity is required for radiation stress-induced epigenetic changes at sites of vascular genes. But its disruption might also dysregulate the global DNA repair response and chromatin integrity.

Further studies will be needed to determine the precise mechanism by which P300 mediates chromatin decompaction and DNA repair in radiated glioma cells.

The net functional outcome of P300 disruption in radiated tumor cells is the inhibition of vascular-like conversion, reduction of tumor growth and enhanced animal survival, indicating that P300 is a potential target for enhancing the effects of radiation. P300 may also play roles in promoting GBM growth outside of the context of radiation. Prior studies have suggested that P300 can be either tumor-suppressive or oncogenic[54]. In our GBM models, we find that P300 plays a pro-tumorigenic role, even in the absence of radiation, as P300-deficient tumors diminish tumor growth in vivo. This effect is unlikely to be mediated by an effect on the production of vascular-like cells by non-irradiated GSC, as we did not observe any significant impact of P300 inhibition or knockdown on vascular gene expression. The general function P300 in gliomagenesis remains to be elucidated.

In summary, the findings presented here provide avenues to address radiation-induced phenotypic plasticity and resulting resistance in GBM. Whether similar processes occur in other radiation-treated cancers remains to be seen. Early studies on epigenetic mechanisms in mediating therapeutic resistance in GBM have largely focused on drugs that target DNA methylation, chromatin remodeling or histone acetylation using HDAC inhibitors[55]. However, our findings highlight HAT as key epigenetic drivers mediating pro-tumorigenic programs and adaptive resistance in GBM, and the use of small molecule HAT inhibitors for treating cancer.

## Methods
All experiments in this study comply with relevant ethical regulations and has been approved by the UCLA Institutional Review Board. All animal studies were performed according to approved protocols by the institutional animal care and use committee at UCLA. All patient samples were de-identified and collected under informed consent and with the approval of UCLA Medical Institutional Review Board.

### Experimental cell lines, culture and treatment
All patient-derived gliomasphere lines were established in our laboratory, and cultured as previously described[56]. Experiments were performed only with lines that were cultured for <20 passages since their initial establishment, and tested negative for mycoplasma contamination. Murine GBM cell line (NRAS G12V-shp53-shATRX-IDH1 wildtype) was obtained from Dr. Maria Castro and cultured in the same media as patient-derived gliomaspheres. Human Umbilical Vein endothelial cells (HUVEC) were purchased from Sciencell (#8000) and maintained in endothelial cell media (R&D systems, CCM027). FACS-sorted, noninduced and radiation-induced GEC and GPC were maintained in endothelial cell media (R&D systems, CCM027) and pericyte media (Sciencell, Cat # 1201), respectively. GBM lines were treated with the HAT inhibitor, C646 (Selleckchem, S7152) at a concentration of 10 μM. Diphtheria Toxin, DT was added to GBM lines at a concentration of 5-10 nM to deplete cells expressing DTR (EMD Millipore Sigma, 322326).

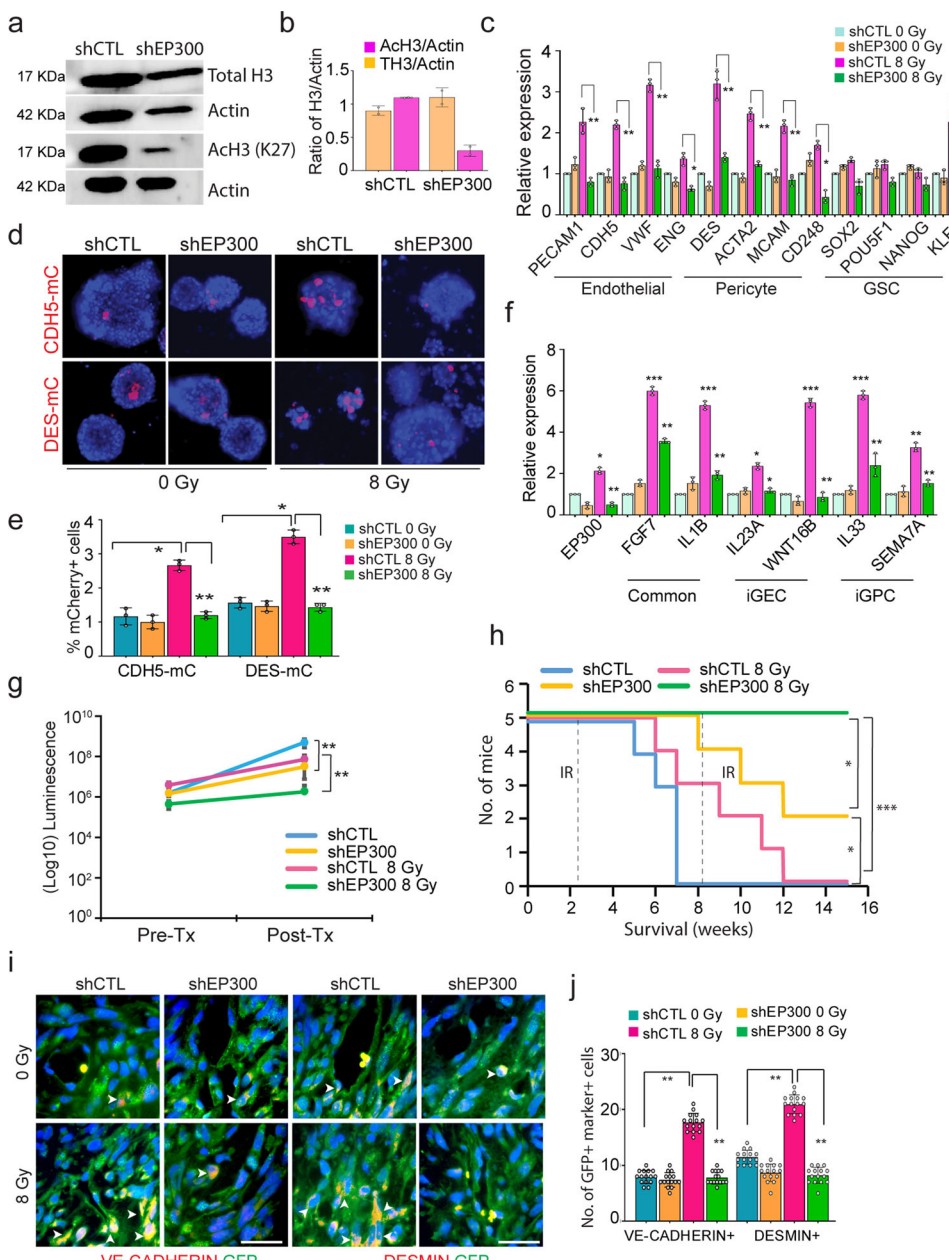

**Fig. 7 | EP300-deficient glioma cells show reduced vascular-like conversion and tumor growth post-radiation treatment. a**, **b** Immunoblots of total and AcH3 in EP300-deficient (shEP300_A) and control (shCTL) cells. Graphs shows quantitation of protein levels in each condition. **c** Relative expression of endothelial, pericyte and stem genes in EP300-deficient and control cells postradiation. $N = 3$ biological replicates. * and ** indicates $p < 0.05$ and $p < 0.005$, one-way ANOVA and post hoc t-test. **d**, **e** mCherry expression in EP300-deficient and control cells post-radiation. Flow-cytometric quantitation of mCherry+ cells in each group. Error bars represent mean ± SD. $N = 3$ biological replicates, * indicates $p < 0.05$ and ** $p < 0.005$, one-way ANOVA, post hoc t-test. **f** Relative expression of iGEC- and iGPC-enriched growth factors in EP300-deficient and control cells pre- and postradiation. Error bars represent mean ± SD. $N = 3$ biological replicates. *, ** and *** indicates $p < 0.05$, $p < 0.005$ and $p < 0.0005$, one-way ANOVA and post hoc t-test. **g** Quantitation of tumor growth pre- (Pre-Tx) and postradiation (Post-Tx) treatment. $N = 5$ mice, ** indicates $p < 0.005$, one-way ANOVA, post hoc t-test. **h** Kaplan-meier survival curve of mice pre- (Pre-Tx) and post-radiation treatment (Post-Tx). $N = 5$ mice, * and *** indicates $p < 0.05$, and $p < 0.0005$, Log-rank test. **i, j** Immunostaining of VE-CADHERIN and DESMIN in GFP + tumor cells. Arrows point to marker+ GFP + cells. Scale bars, 50μm. Quantitation of GFP + marker+ cells in tumor mass. Error bars represent mean ± SD. $N = 3$ mice, ** indicates $p < 0.005$, one-way ANOVA.

## Animal strains, intracranial transplantation, treatments and imaging

Studies did not discriminate sex, and both male and females were used. Strains: 8 to 12-week-old NOD-SCID gamma null (NSG) mice (NOD.Cg-*Prkdc*$^{scid}$ *Il2rg*$^{tm1Wjl}$/SzJ Jackson Laboratory, 00557) were used to generate tumors from a patient-derived GBM line HK_408. C57BL6 (Jackson Laboratory, 000664) were used to transplant a murine IDH1-wt GBM model (wild-type IDH1 (NRAS G12V-shp53-shATRX) as described previously[25]. $5 \times 10^4$ tumor cells containing a firefly-luciferase-GFP

lentiviral construct were injected intracranially into the neostriatum in mice. For assessing tumor initiation potential of CD133 + and CD133-fractions, $1 \times 10^3$ cells injected intracranially. Co-transplantation experiments were performed at a ratio of 1:1 (tumor: vascular-like cells) $5 \times 10^4$ cells per condition. Treatments: For GCV (Sellechckem, S1878) treatment, animals were injected by i.p at a dose of 80 mg/kg of body weight of mice every day for a week, and for DT (EMD Millipore Sigma, 322326) treatment, animals were given i.p injections at a dose of 5 μg/kg of body weight of mice every 2 days over a week. Imaging:

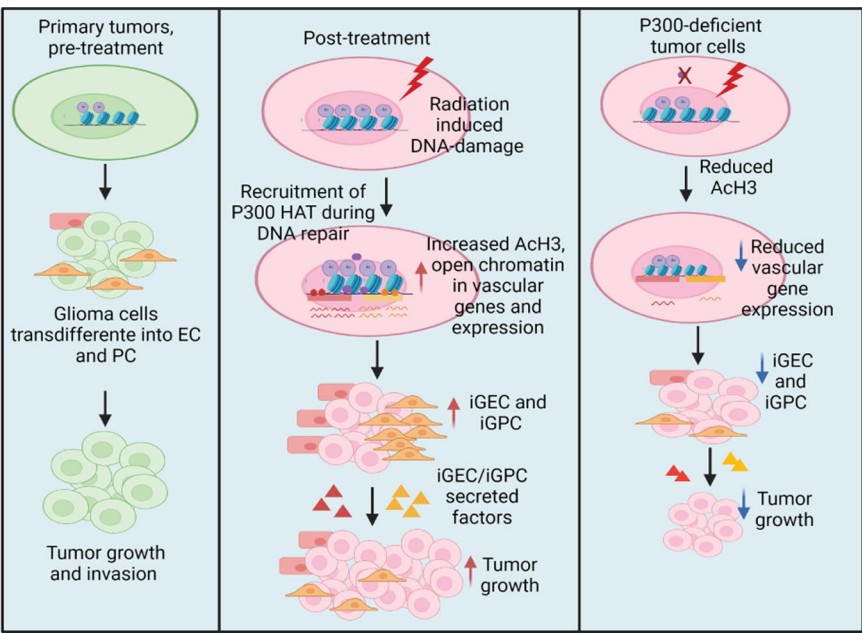

**Fig. 8 |** Model illustrating radiation-induced vascular-like phenotype conversion in glioma stem- and tumor cells via a p300-dependent mechanism.

Tumor growth was monitored 2- and 4 weeks after transplantation by measuring luciferase activity using IVIS Lumina II bioluminescence imaging. ROIs were selected to encompass the tumor area and radiance was used as a measure of tumor burden. Survival: Mice were deemed sick and euthanized when they became symptomatic (i.e. lethargy, decreased activity, dehydration (decreased skin turgor and weight loss>10%), significant skin or fur changes, or interference with any bodily function) for survival experiments.

### Irradiation of gliomaspheres and animals

Cultured cells were irradiated at room temperature using an experimental X-ray irradiator (Gulmay Medical Inc. Atlanta, GA) at a dose rate of 5.519 Gy/min for the time required to apply a prescribed dose. The X-ray beam was operated at 300 kV and hardened using a 4 mm Be, a 3 mm Al, and a 1.5 mm Cu filter and calibrated using NIST-traceable dosimetry. Tumor-bearing mice were irradiated at a single dose of 8 Gy using an image-guided small animal irradiator (X-RAD SmART, Precision X-Ray, North Branford, CT) with an integrated cone beam CT (60 kVp, 1 mA) and a bioluminescence-imaging unit as described previously (Bhat et al. 2020). Individual treatment plans were calculated for each animal using the SmART-Plan treatment planning software (Precision X-Ray). Radiation treatment was applied using a 5×5 mm collimator from a lateral field.

### Single-cell RNA sequencing and analysis

Preparation of single-cell suspensions. Gliomaspheres: Single-cell suspensions of cultured control (0 Gy) and 2- and 7-day radiated (8 Gy) gliomaspheres ($N = 3$ replicates per condition) were pooled, and prepared on the same day using 10x Genomics Chromium Single Cell 3′ Reagent Kits v3 according to the manufacturer's protocol. Tumor xenografts: Control and radiated tumors ($N = 3$ mice per group) were enzymatically dissociated with Collagenase II, Trypsin and DNAse to generate cell suspensions. Contaminating mouse cells were removed using MACS mouse cell depletion kit (Miltenyi Biotec, 130-104-694). Viable GFP + tumor cells were FACS sorted and collected for preparing single-cell suspensions using 10x Genomics Chromium Single Cell 3′ Reagent Kits v3. Quality control and sequencing: Quantity and quality of cDNA were assessed by Agilent 2100 expert High Sensitivity DNA Assay. cDNA samples were sequenced on 1 lane of NovaSeq 6000 S2 flowcell. Reads were mapped to HumanGRCh38 genome using Cell

Ranger v.3.0.2 (10X Genomics). Gliomaspheres: More than 600 million reads were obtained for each sample. An average number of genes detected was 5875 (SE ± 357). Confident read mapping rates were 81.2-87.4% with over 86.8% of reads in cells. Filtering genes and cells: Seurat package v3.1.1 (https://satijalab.org/seurat/) was used to do analysis. For each condition, genes expressing in less than 5 cells were removed. Cells with number of features less than 500 were excluded. PCA was used for dimension reduction with top 5000 most variable genes. Raw count data were normalized using regularized negative binomial regression with SCTransformation, batch correction was done with Seurat's integration method using Canonical Correlation Analysis (CCA) with 75 dimensions. Cell clustering is done using Shared Nearest Neighbor (SNN) Graph method. Markers were identified by Seurat's FindMarkers function using Wilcox Rank Sum test with minimum 25% cells expressing. Expressed gene list sorted by directional log10(p-Values) were used as input for enrichment analysis by GSEA (http://software.broadinstitute.org/gsea/index.jsp). Peak enrichment score of each gene set is determined by walking down the sorted gene list, which reflects the skewedness of the distribution of gene set in the provided gene list. Normalized enrichment score (NES) was used as the primary statistics for examining gene signature enrichment. Each cell cluster was annotated by a combination of the following methods 1) canonical marker expression 2) Gene Set Enrichment Analysis (GSEA) of cluster-specific markers determined by Find Markers function in Seurat and 3) Co-expression modules obtained by Louvain community detection clustering method. For GSEA, MSigDB ver7.0 was used as reference. Trajectory analysis was done with monocle 3 (version 0.2.1.3) by converting Seurat object to monocle. Gene modules are determined with significant genes by Molan's I test (q < 0.05). Tumor xenografts: Over 200 million reads were obtained for each sample. Average number of genes detected was 6362 (GBM_TX 0 Gy) and 4824 (GBM_TX 8 Gy), and confident read mapping rates were over 76.5-78% with over 88% of reads in cells. In vivo single-cell dataset was similarly analyzed to in vitro dataset except the following: the two in vivo expression matrices from CellRanger output were merged, then low expression genes present in less than 5 cells and cells with less than 500 or more than 8000 features, as well as mitochondrial contents greater than 5 percent were filtered. Mitochondrial and ribosomal genes were excluded for downstream analysis. Filtered expression data were individually normalized using SCTransform in Seurat

functions. PCA was used for dimension reduction with top 3000 most variable genes. The batch effect was corrected using harmony.

## Bulk RNA-sequencing and analysis

RNA extraction from gliomaspheres ($N = 2$ to 3 replicates per condition) was performed using Qiagen RNeasy microkit. RNA quality was assessed using Bioanalyzer and only samples with a RIN score >8.0 were sequenced. RNA samples were pooled and barcoded, and libraries were prepared using TruSeq Stranded RNA (100 ng) + Ribozero Gold. Paired-end 2X75bp reads were aligned to the human reference genome (GRCh38.p3) using the STAR spliced read aligner (v 2.3.0e). Total counts of read fragments aligned to known gene regions within the human hg38 refSeq reference annotation was used as the basis for the quantification of gene expression. Differentially expressed genes were identified using EdgeR Bioconductor R-package, which are then considered and ranked based on False Discovery Rate (FDR Bejamini Hochberg adjusted p-values of ≤ 0.01(not 0.1?)). Gene Set Enrichment Analysis (GSEA) was carried out to determine the gene signatures differentially regulated in control and radiated cells and represented as heatmaps. R-Package V.3.2.5 (The R project for Statistical Computing, https://www.r-project.org/) was used to generate the PCA plots and heatmaps.

## Quantitative RT-PCR

RNA was isolated with RNeasy Micro or Mini Kit (QIAGEN), and then used for first-strand cDNA synthesis using random primers and Superscript Reverse Transcriptase (Invitrogen). qRT–PCR was performed using Power SYBR Green PCR Master Mix (Applied Biosystems). The relative expression of genes was normalized using 18srRNA as the housekeeping gene. All experiments were repeated 3 times with 3 replicates per condition, and data is represented as mean ± SD. Primers are listed in the Supplementary Table 2.

## Lentivirus-transduction in tumor cells

Lentiviral vectors (pLV(EXP)-CDH5-mCherry:T2A:Puro, pLV(EXP)-DES-mCherry:T2A:puro, pLv(EXP)-puro-DES-GFP, pLV(EXP)- mCherry:T2A-puro- CDH5-HSV-TK, pLV(EXP)- mCherry:T2A:Puro- DES-HSV-TK-, pLV(EXP)- mCherry:T2A:puro- CDH5-DTR and pLV(EXP)-mCherry:T2A:puro-DES-DTR) were designed and purchased from Vector builder. CDH5, ANGPT1, EP300 and Scrambled-siRNA constructs were purchased from Abmgood (cat# 156710910395 (CDH5-shRNA set), 118940910395 (ANGPT1-shRNA set) and 193330910395 (EP300-shRNA set). Gliomaspheres were transduced with indicated viruses for 48 hours and selected with puromycin (EMD Millipore Sigma, P9620) for 72 hours. Reporter gene expression and knockdown was validated by immunofluorescence, quantitative RT-PCR or immunoblotting in target cells.

## Flow cytometry and FACS sorting

Cells were harvested and suspended in ice-cold PBS with 1% BSA and 2 mM EDTA. After incubation with FcR Blocking Reagent (Miltenyi Biotec), cells were stained by fluorescently conjugated antibodies and incubated for 10 min in the dark in the refrigerator (2 – 8 °C). Antibodies include CD31-PE, CD144-APC, CD146-PE CD248-APC, IgG-APC and IgG-PE from Miltenyi Biotec (antibody information and dilution used is listed in the supplementary table 1). DAPI staining was used for dead cell exclusion. The stained cells or GFP and mCherry-labeled cells were analyzed in a BD Fortessa analyzer. FACS sorting was performed using the BD FACSAria cell sorter. Data were analyzed using FlowJo v10.8 software (https://www.flowjo.com/). A representative gating strategy for sorting mCherry+ cells is included in the source data.

## Immunofluorescence (IF) staining

For IF staining, 5μm FFPE brain sections were incubated with primary antibodies overnight at 4 °C after deparaffinization, rehydration,

antigen retrieval and blocking in PBS with 2% BSA. Sections were then incubated with species-appropriate goat/donkey secondary antibodies coupled to AlexaFluor dyes (488 and 568, Invitrogen) and Hoechst dye for nuclear staining for 2 hrs at RT. VECTASHIELD (Vector Laboratories) was used to mount coverslips. Primary and secondary antibodies used for immunostaining are listed in Supplementary Table 1. Slides were imaged using Leica LAS X or EVOS microscope, and quantification was performed using ImageJ (https://imagej.nih.gov/ij/). For quantitation of stained sections, marker+ GFP + tumor cells in the tumor mass or per blood vessel (BV) spanning 0.04 mm² area per section was counted in 5 random sections per tumor and at least 3 tumors per condition in a blinded fashion. Data is represented as mean ± SD in the graphs.

## Matrigel tube formation and matrigel plug in vivo angiogenesis assay

Tubular network formation was assessed by growth factor reduced (GFR) Matrigel assay kit (CB40230C, Fisher Scientific/Corning) in three-dimensional (3D) culture according to the manufacturer's instructions. Briefly, tumor cells were harvested 7days post-radiation and cultured on GFR matrigel for 16-24 hours. Images were obtained at 20x magnification on EVOS microscope and number of branch points was quantitated manually from 5 random fields per well. Matrigel plug in vivo angiogenesis assay: $5 \times 10^5$ labelled sorted GFP + tumor and mCherry+ GFP + transdifferentiated cells from control and radiated groups were embedded in GFR matrigel. Matrigel containing cells were injected subcutaneously into mice ($N = 5$ per condition) and allowed to solidify to form a plug. 7 days after implantation, plugs were harvested and fixed in 4% PFA. Sections were stained with Hematoxylin & Eosin stain and Masson's Trichrome to label the basement membrane and blood vessels. Images were obtained at 20x magnification using EVOS microscope. Quantitation of vessel density was performed using Image J plugin for Vessel analysis. Percent vessel density in each condition represents mean ± SD.

## DiI-Ac-LDL uptake assay

Control and radiated gliomasphere cultures were incubated with DiI-Ac-LDL (Cell applications, 022k) for 24 hours. Cells were fixed, and stained with VE-CADHERIN and imaged using EVOS microscope. Quantitation of number of DiI-Ac-LDL + cells was performed by manually counting the double-positive cells from 5 random fields per condition in a blinded fashion. Data is represented as mean ± SD in the graphs, and p-values are derived from 3 independent experiments.

## Limiting dilution assay

For assessment of self-renewal of sorted CD133 + and CD133- fractions, cells were plated at a density of 1, 5, 10, 25 and 50 cells per well (10 wells for each density). Cells were maintained for 10 days before sphere formation was evaluated. Spheres larger than 10 cells in diameter were considered for analysis. Numbers represent stem cell frequency as calculated using the Walter and Eliza Hall Institute Bioinformatics Division ELDA analyzer http://bioinf.wehi.edu.au/software/elda/.

## Cell proliferation assay

GBM cells were plated at a density of 5000 cells per well in 96-well plates. Proliferation was assessed 3 days after treatment with inhibitors or conditioned media using CellTiter-Glo® Luminescent Cell Viability Assay (Fisher Scientific, PRG9242). Recombinant growth factors were added at a concentration of 50–100 ng/ml (hFGF7, Peprotech, 100-19-10ug), hIL1B (Thermofisher scientific, PHC0814), hIL23A (R&D systems, 1290-IL-010/CF), hIL33 (Thermofisher Scientific, 34-8539-63), hWNT16B (R&D systems, 7790-WN-025/CF) and hSEMA7A (Fisher Scientific, 2068S7050). The luminescence signal was measured in a luminometer, and readings were taken on Day 0 of plating and at Day 3 after treatment to normalize for plating density. $N = 3$ replicate per

condition, and repeated 3 independent times. Error bars represent mean ± SD.

## ATAC-sequencing and analysis

Control and radiated gliomaspheres ($N = 3$ replicates per group, and repeated at least 2-3 independent times) were processed according to the protocol, Buenrostro JD et al., 2013[57]. ATAC-Libraries from each condition was sequenced on NextSeq 500 High Output Kit v2 (150 cycle, FC-404-2002, Illumina). Alignment of reads was carried out using the Burrows-Wheeler Aligner mem using hg19 assembly. Peak calling was performed using MACS2 (with parameter setting –nomodel –shift 75), and differential peak analysis using featureCount and DESeq2 (default setting). Motif analysis and peak annotation was done using HOMER and GO analysis using HOMER (http://homer.ucsd.edu/homer/) and EnrichR (https://maayanlab.cloud/Enrichr/). UCSC Genome Browser was used to determine whether open regions displayed H3K27Ac and conserved TF binding sites. Integrated Genome viewer (IGV) was used to represent the peaks/open regions.

## Chromatin Immunoprecipitation (ChIP)-quantitative RT-PCR

Chromatin immunoprecipitations were performed using SimpleChIP® Enzymatic Chromatin IP Kit (Magnetic beads, Cell Signaling Technology, #9005 S #) according to manufacturer's instructions. Briefly, formaldehyde cross-linked chromatin from each sample ($N = 3$ per condition, and repeated 3 independent times) is enzymatically digested using Micrococcal Nuclease, followed by sonication to obtain chromatin fragments. Fragmented chromatin was incubated with anti-rabbit IgG or anti-rabbit H3K27Ac antibodies for immunoprecipitation with Protein-G magnetic beads. After reverse-crosslinking, DNA is purified and analyzed by quantitative RT-PCR. Primers for human RPL30 was provided with the kit. Primers for CDH5 and ANGPT1 were designed specifically for open regions identified from ATAC-sequencing in radiated cells. CDH5 Forward primer: CATAAAAGTCCTTCCCATGTTGC, CDH5 Reverse primer: TGGCAATGAAGAGTAGTCCCAA. ANGPT1 Forward primer: CGACAGTTGCCATCGTGTTC and ANGPT1 Reverse primer: TTTCCTCGCTGCCATTCTGA were custom synthesized from Integrated DNA technologies. Quantitation of DNA was performed using RT-PCR and fold enrichment was normalized to input control.

## Immunoblotting

Cells were harvested, washed and lysed in RIPA buffer with protease inhibitor cocktail and centrifuged at 13,000 rpm for 10 min. Protein concentration in each sample was determined by Bradford assay using BSA as a standard. 15ug of protein was used for each sample. Protein lysates were subjected to SDS-PAGE on 4%–12% gradient polyacrylamide gel (Thermo-fischer Scientific), transferred onto nitrocellulose membranes and incubated with primary antibodies, washed, and probed with HRP-conjugated secondary antibodies. Quantitation of protein levels was performed in ImageJ by normalizing to loading control, β-actin. Data is represented as mean ± SD, and derived from 2 to 3 independent experiments. Antibodies are listed in supplementary table 1. Full scans of representative blots are provided in the source data file.

## Statistics and Reproducibility

All data are expressed as mean ± SD. Quantification of qRT-PCR, cell-based proliferation assays, immunoblotting are representative of at least three independent experiments unless otherwise states. $P$ values less than 0.05 were considered to be significant, and were calculated in Graph Pad Prism 9.0 using unpaired two-tailed Student t-test and one-way ANOVA for multiple comparison with Bonferroni correction, followed by post-hoc t-test. Log-rank analysis was used to determine the statistical significance of Kaplan-Meier survival curves. No samples, mice or data points were excluded from the analysis reported in this study. R-package was used for statistical analysis of sequencing experiments. Schematics used in the figures were generated in Biorender (https://biorender.com/).

## Reporting summary

Further information on research design is available in the Nature Research Reporting Summary linked to this article.

## Data availability

All sequencing data has been submitted to Gene Expression Omnibus, and are available with the Accession number GSE207808. All other quantitative data from this study can be obtained in the source data file provided with this paper or made available upon request to the corresponding author. https://www.ncbi.nlm.nih.gov/geo/query/acc.cgi?acc=GSE207808 Source data are provided with this paper.

## Code availability

No custom code was used in this study. Any code used for generating sequencing data will be made available upon request to the corresponding author.

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

## Acknowledgements

The authors thank the UCLA pathology, flow cytometry, TCGB and UNGC sequencing cores for their technical assistance with support from the Jonsson Comprehensive Cancer Center P30CA016042 and Dr. Paul Mischel for helpful discussions on the manuscript. This work was funded by Broad Stem cell postdoctoral fellowship (SDM), UCLA, The Dr. Miriam and Sheldon G. Adelson Medical Research Foundation (RK, SAG, DHG, HIK), the UCLA SPORE in Brain Cancer P50 CA211015-01A1 (FP, HIK), NIH RO1NS121617 (HK), NIH R01HL149687 (AD), NIH R01CA200234 (FP) and the UCLA Intellectual and Developmental Disability Research Center P50 HD103557 (HIK).

## Author contributions

S.M. conceptualized and performed the experiments, analyzed the data and wrote the manuscript. S.M., P.N., R.P., M.J., N.V., M.C.C., A.G.A., R.G., A.P. and Q.W. conducted the experiments. R.K, F.G. and Y.Q. performed the computational analysis. T.B., D.G., M.C., and P.L. provided resources, reagents and contributed to experimental discussions. F.P., A.D., J.H., and S.G. contributed to experimental design and data analysis. H.K. supervised the work and edited the manuscript.

## Competing interests

The authors declare no competing interests.
