## [Peer Review File · Nature Communications]

REVIEWER COMMENTS

Reviewer #1 (Remarks to the Author):

The authors of this study have employed single-cell and whole transcriptomic analyses to uncover that radiation-stress induces a dynamic shift in functional states of glioma cells allowing for acquisition of vascular endothelial-like and pericyte-like cell phenotypes. Vascular-like cells provide trophic support to promote proliferation of tumor cells, and their selective depletion results in reduced tumor growth post-treatment in vivo. The acquisition of vascular-like phenotype is driven by increased chromatin accessibility and H3K27 acetylation in specific vascular gene regions allowing for their increased expression post-treatment. Blocking P300 histone acetyltransferase activity reverses the epigenetic changes induced by radiation, and inhibits the adaptive conversion of glioma stem cells into vascular-like cells and tumor growth. REMARKS. This is a very interesting and original work. I suggest to the Authors to further discuss the important role of epigenetic changes in tumor progression in human glioblastoma and the alternative mode of tumor vascularization, including vascular co-option, operating in glioblastoma and their role in the development of resistance to therapy.

Reviewer #2(Remarks to the Author):

The authors report that subpopulations of glioma cells are induced to express vascular lineage genes upon challenge with radiation. The vascular-like potential appears to reside within a CD133+ fraction. These vascular-like glioma cells only rarely incorporate into vascular structures. Instead, the authors suggest that the vascular-like cells provide trophic support to glioma cells during tumor rebound following radiation. This trophic support is suggested to come in the form of secreted ligands including, for example, SEMA7A. Finally, the authors use various assays to quantify open/closed chromatin around regions of vascular-lineage genes and find that radiation results in increased chromatin accessibility and H3K27 acetylation; blocking p300 (a HAT) reverses the epigenetic changes induced by radiation and inhibits the conversion of glioma cells into vascular like cells.

Major points (in no particular order)

This is an interesting study that adds to the existing body of literature suggesting a vascular-like potential of glioma cells (i.e. an ability of glioma cells to convert to endothelial-like or pericyte-like cells and form vascular structures alone or in collaboration with existing vasculature). However, in reality, the authors find little evidence for direct incorporation of these induced, vascular-like glioma cells into functional vascular structures which contradicts some previous reports. These data should be expanded upon more thoroughly to include assays to test for functionality of any glioma-derived vascular structures, and higher quality images of the structures themselves. Many of the images are of poor quality and it is difficult to tell what is being emphasized.

The increase in vascular-lineage genes induced by radiation is rather modest in most assays. Moreover, many of the images that putatively show expression of endothelial or pericyte markers in glioma cells are of poor quality and are not convincing. Examples include, but are not limited to, fig 3C and data showing Ac-LDL uptake (actually a read-out for expression of a scavenger receptor that is typically expressed by endothelial cells and macrophages).

It is not clear why the authors have pursued the chromatin accessibility angle around vascular-lineage genes (e.g. *Cdh5* and *Angpt1*) when there are no clear ties to the function of these genes in the glioma setting post radiation according to the authors. The authors own data led them to examine trophic effects of these vascular-like glioma cells. Yet pursuit of the function of these trophic factors in vivo is not well-developed. For example, the authors do not mention whether P300 blockade has any impact on the trophic factors that are apparently responsible for tumor regrowth after radiation or whether depletion of sub-populations of glioma cells also reduces the presence of the trophic factors. The apparent disconnect between the function of the vascular-

lineage genes in glioma cells, the trophic factors, chromatin accessibility, and glioma rebound after radiation is a confusing aspect of these work that makes the overall theme/message of the manuscript hard to appreciate.

A major issue with the paper is lack of clarity or information or consistency on the experimental aspects of the work (number of repeats, number of mice, etc). In some cases, the statistical test chosen appears to be incorrect or needs clarification (see below). This has made it challenging to judge the merits of the manuscript overall and it is not clear how robust the data are in total.

Fig 1

It is not clear how many samples were analyzed (how many repeats?). Is this just one sample being assayed? The hallmark angiogenesis genes are going down over time? Also, to this reviewer, there is a lot of extraneous information in this figure that make it hard to follow overall; examples include numerous GO-terms with no apparent connection to the study at large (or at least this has not been well-explained to the reader).

Fig 2

The images in panels H and K could be improved with confocal images of the individual channels. Functional parameters of the blood vessels in question should also be assessed. Again (and this is a theme in almost every figure), information on the number of experimental repeats (technical versus biological) are lacking in most panels. Just one example is panel D. All data should be at least $n=3$ independent biological replicates and individual data points should be shown. Panels I and J have only $n=2$. Can appropriate statistics be done on $n=2$?

Fig 3

Ac-LDL images are not convincing. It is not clear what panel F is showing. Again, how many repeats were done for the experiments? How many mice per group? How many sections were analyzed? How were these data quantified in an unbiased way? It is hard to assess whether any of these data are rigorous or not. The Matrigel tube forming assay is not really a robust indicator of vascular-like potential. It is not clear whether a patent lumen is present. Many cells, even fibroblasts, will self-organize into transient "tube-like" structures when plated at certain densities on Matrigel or other ECM, but this is no indication that fibroblasts can form bona fide blood vessels.

Fig 4

Data in B and D appear modest. For example, the increase in trophic factors is quite minimal. Only RNA is assayed. How about protein levels? Do the levels of these factors reach relevant physiologic levels/activities in vivo? Again, how many repeats were done here? Three biological replicates in duplicate or triplicate should be done. Do the glioma cells express the receptors for these trophic factors? In its present form, links between an increase in these trophic factors and glioma survival are correlative. It is hard to follow the data in panel F and G. Again, how many repeats? Is this $n=1$ experiment with $n=5$ mice? This would not appear to be robust enough to assess whether the results are rigorous.

Fig 6

Panel B is $n=2$? How can appropriate statistics be carried out? Why was a t-test used in C? These data should be assessed using an ANOVA since there are multiple variables/comparisons. The same is true for other graphs in this figure. Individual data points should be shown as well (mean value \pm STD for minimum of 3 biological replicates ideally assayed in duplicate or triplicate).

Fig 7

Are the trophic factors identified in this study also impacted by the p300 inhibitor? How many mice in G? How many repeats?

Fig S1

How have the authors distinguished bona fide endothelial cells from glioma-derived endothelial-like cells or pericytes in the scRNA-seq analysis? How was the purity of these samples assayed? Panel J is not convincing, nor is it quantified. Some of this staining almost looks non-specific to this reviewer w/o adequate negative controls or high quality images showing membrane staining of Cdh5 or Pecam.

Fig S2

How many biological replicates? Individual data points should be shown for all graphical data. In panel N there is no legend callout. What is this showing?

Fig S3

Ac-LDL images are not convincing. DI-Ac-LDL uptake by endothelial cells typically has a punctate appearance by microscopy.

Fig S4

How many mice? How many biological replicates.

Fig S5

In panel B, none of these pathways appear to support vascular-like features of glioma cells – is that what is meant to be shown here? Panel D is not convincing. The results are very modest and based on the images, it is not clear how these data were quantified in a rigorous or unbiased way. Were ROIs chosen? Were multiple tumors imaged? How were these data normalized to account for differences in tumor size?

Fig S6

For some panels, how many biological/technical replicates?

Reviewer #3 (Remarks to the Author):

Muthukrishnan et al. treated gliomaspheres and glioma xenografts with ionizing radiation (IR) and profiled specimens from those experiments with single-cell and bulk transcriptomics and immunofluorescence. These studies were complemented with in vivo angiogenesis, in vivo treatment with histone acetyltransferase inhibitor assays, and several other in vitro functional assays. The authors claim that IR causes a shift to a mesenchymal phenotype in vitro and in vivo, that IR induces endothelial- and pericyte-like phenotypes in glioma stem cells. They claim that depleting cells with a vascular tumor phenotype enhances survival, that mechanistically vascular conversion is regulated by histone acetylation of vascular genes and that targeting histone acetyltransferase reduces vascular conversion. The study is timely and addresses several current topics in glioma biology. The approach is thoughtful, the experiments are generally well controlled, the bioinformatics analysis is rigorously done. In many ways, the manuscript is largely complete. There are several minor to moderate weaknesses that diminish enthusiasm slightly. The authors may want to address the following points:

1. The reporting on the glioma-spheres and xenografts should be improved. This ties into the role of potential epistatic effects and generalizability of the authors' conclusions that appear to be based on a single glioma culture. There is considerable inter-patient heterogeneity in GBM and it appears that the authors have used cultures from a single patient sample for all figures except figure 3. How many distinct patient samples were used to derive the cells for scRNA-seq/RNA-seq/IF? What is the genotype of these cultures, do they have EGFRviii, NF1, P53 or other mutations that would affect how we interpret mesenchymal signatures or IR responses? Validating some of the basic

correlations in additional cultures with different genetics and phenotypes would strengthen the manuscript.

2. The authors show that ionizing radiation induces mesenchymal transition and vascular conversion. Prior studies implicated TNF-alpha in mesenchymal transition following IR, with NFKB activation being essential. Are these pathways, and in general prior studies of IR-induced mesenchymal transition, supported by the authors' data? Do the authors have any evidence for the mechanism of IR induced vascular conversion?

3. In the authors' data, is vascular conversion strictly tied to mesenchymal transition? Do most of the vascular cells in their glioma-spheres and xenografts express markers of mesenchymal cells?

4. Evidence suggests that the mesenchymal phenotype is inducible and that mesenchymal cells can convert to other phenotypes. So, the mesenchymal state may represent a transient state, is this the case for vascular-like states? Blocking P300 HAT inhibits vascular conversion in the first place, but can the vascular phenotype be regressed?

5. When only one shRNA is used per target, as in Figure 7, it is natural to worry about off-target effects.

6. The survival analysis in Figure 4G is somewhat surprising, in that 1) co-injection with vascular cells and no radiation does better than unsorted cells. 2) co-injection with vascular cells and no radiation does the same as vascular-depleted (mC-) cells without radiation. Do the authors have a comment on this?

7. In Figure 4H-J the authors report tumor growth after 6 weeks, but not overall survival?

8. There is a strong preclinical rationale for HDAC inhibition in GBM. This is essentially the opposite of blocking HAT or at least antagonistic. I would ask the authors to address this in the discussion and I am curious to hear their comment on this.

Reviewer #4 (Remarks to the Author):

Glioblastoma is a challenging disease to treat, in part because standard radiation and chemotherapy are insufficient to eradicate the tumor cells. In fact recurrence after radiation therapy, is not well understood. The authors who jest that radiation stress and this is a dynamic shift in glioma cells to acquire vascular and parasite like cell phenotypes. They suggest that this allows for trophic support and proliferation of tumor cells the selective depletion of which reduces tumor growth. They suggest that blocking P300 histone acetyltransferase activity reverses these epigenetic changes induced by radiation. Therapeutic targeting of P300 the author suggest will inhibit the therapy induced adaptive response.

Strengths of the projet include the focus on GSC that can convert under certain stress conditions (eg radiation) and may ultimately be responsible for recurrence, in particular the endothelial and parasite transition. the breadth of information across single cell in bulk transcriptome over many patient-derived glioma sphere cultures, orthotopic xenografting, further support the strength of their results. Overall the thoughtful composition of experiments with detailed controls and biostatistical analysis highlight the significance of these results. the main limitation includes reflection on primary human samples or correlation with patient survival or grade in response to radiation

The contribution of radiation-induced phenotypic plasticity and GBM was mainly performed using single cell RNA sequencing of primary glioma sphere cell lines. How do these specific data then reflect on expression or ATACseq analysis on recurrent (radiation resistant) glioblastoma? The initial data both in the tumor spheres as well as in NSG mice lack the contribution of the immune component. Is there any suggestion either in data from this group or otherwise that this could confound your results? Likewise, patient derived glioma spheres represent a subset of selected cells that can be maintained or proliferate under these conditions. The incorporation of a single syngeneic mouse model validation experiment is helpful but these concerns remain. Given that P300 disruption could dysregulate other mechanisms of DNA repair after radiation, the precise mechanism in glioma remains unknown. C646, a selective small molecule inhibitor of P300 is a known radiosensitizer in other cancers.

RESPONSE TO REVIEWERS' COMMENTS

Reviewer #1 (Remarks to the Author):

The authors of this study have employed single-cell and whole transcriptomic analyses to uncover that radiation-stress induces a dynamic shift in functional states of glioma cells allowing for acquisition of vascular endothelial-like and pericyte-like cell phenotypes. Vascular-like cells provide trophic support to promote proliferation of tumor cells, and their selective depletion results in reduced tumor growth post-treatment in vivo. The acquisition of vascular-like phenotype is driven by increased chromatin accessibility and H3K27 acetylation in specific vascular gene regions allowing for their increased expression post-treatment. Blocking P300 histone acetyltransferase activity reverses the epigenetic changes induced by radiation, and inhibits the adaptive conversion of glioma stem cells into vascular-like cells and tumor growth.

REMARKS.

This is a very interesting and original work.

I suggest to the Authors to further discuss the important role of epigenetic changes in tumor progression in human glioblastoma and the alternative mode of tumor vascularization, including vascular co-option, operating in glioblastoma and their role in the development of resistance to therapy.

Response: As suggested by the reviewer, we have included the role of vascular co-option and vascular mimicry in mediating resistance to therapy and also discussed the potential role of epigenetic changes in tumor progression in the discussion section as follows:

(Page 24, lines 595-600) "Several mechanisms of neovascularization have been identified in GBM in addition to angiogenesis, and include vasculogenesis, vessel co-option, vascular mimicry and endothelial- and pericyte-transdifferentiation. These alternative neovascularization mechanisms are induced in response to anti-angiogenic therapy or by standard treatments of radiation and TMZ and contribute to therapeutic resistance (38,39).

(Page 26, 659-663) "Prior studies on epigenetic mechanisms in mediating therapeutic resistance in GBM have largely focused on drugs that target DNA methylation, chromatin remodeling or histone acetylation using HDAC inhibitors (55)".

Reviewer #2(Remarks to the Author):

The authors report that subpopulations of glioma cells are induced to express vascular lineage genes upon challenge with radiation. The vascular-like potential appears to reside within a CD133+ fraction. These vascular-like glioma cells only rarely incorporate into vascular structures. Instead, the authors suggest that the vascular-like cells provide trophic support to glioma cells during tumor rebound following radiation. This trophic support is suggested to come in the form of secreted ligands including, for example, SEMA7A. Finally, the authors use various assays to quantify open/closed chromatin around regions of vascular-lineage genes and find that radiation results in increased chromatin accessibility and H3K27 acetylation; blocking p300 (a HAT) reverses the epigenetic changes induced by radiation and inhibits the conversion of glioma cells into vascular like cells.

Major points (in no particular order)

1. This is an interesting study that adds to the existing body of literature suggesting a vascular-

like potential of glioma cells (i.e. an ability of glioma cells to convert to endothelial-like or pericyte-like cells and form vascular structures alone or in collaboration with existing vasculature). However, in reality, the authors find little evidence for direct incorporation of these induced, vascular-like glioma cells into functional vascular structures which contradicts some previous reports. These data should be expanded upon more thoroughly to include assays to test for functionality of any glioma-derived vascular structures, and higher quality images of the structures themselves. Many of the images are of poor quality and it is difficult to tell what is being emphasized.

Response: While previous studies did report incorporation of vascular-like cells (predominantly glioma-derived pericytes) into vessels in murine GBM models and xenografts, the glioma-derived endothelial-like cells are seen more rarely in vessels in both xenograft models and human tumors. Our finding from murine GBM model is that only a small percentage of the radiation-induced cells (both EC- and PC-like, 1-2 cells per vessel in a 200 μm^2 area; Figure S2M, O) incorporate into vessels, but the majority of the cells (18-20 cells per 200 μm^2 tumor area) are outside the vessels in the tumor mass Figure 2G). We have discussed our theory that the main function of radiation-induced vascular-like cells is not blood supply or vessel formation, rather providing trophic factors to aid in survival and proliferation of radiated tumor cells. This is not contradictory to prior literature, as induced vs non-induced cells may play different roles in primary versus recurrent GBM.

2. The increase in vascular-lineage genes induced by radiation is rather modest in most assays. Moreover, many of the images that putatively show expression of endothelial or pericyte markers in glioma cells are of poor quality and are not convincing. Examples include, but are not limited to, fig 3C and data showing Ac-LDL uptake (actually a read-out for expression of a scavenger receptor that is typically expressed by endothelial cells and macrophages).

Response: We agree with the reviewer that the vascular-lineage gene induction post-radiation is modest. We believe that this is because only a small percentage of cells (about 5-8%) within the tumor mass express these markers both in vitro as well as in vivo. But this effect is robust, as we see this across many patient-derived GBM lines spanning different subtypes and mutations, and by immunostaining, cell-type specific reporter activity and by gene expression. LDL-uptake assay is one functional assay to show that these cells possess characteristics of normal EC as are matrigel tube formation and matrigel plug assay. We do not claim that these are bonafide EC as these are tumor-derived cells mimicking vascular cells. We have repeated the experiment and added higher magnification images of the LDL uptake as per the reviewer's suggestion as shown below (Fig3C)

3. It is not clear why the authors have pursued the chromatin accessibility angle around vascular-lineage genes (e.g. Cdh5 and Angpt1) when there are no clear ties to the function of these genes in the glioma setting post radiation according to the authors.

Response: Cell fate transitions are generally associated with altered chromatin accessibility and thus we suspected that radiation induced DNA damage would result in chromatin modifications, including alterations in histones and accessibility that would explain the observed cell fates. Because we observed changes in expression of vascular genes, we reasoned that we would observe increased accessibility in the promoters for these genes. Indeed, our ATAC-seq data showed that chromatin was altered in vascular-lineage genes but not for other genes as shown in Figure 5. There is a prior study with regard to CDH5 function in glioma (Mao X et al., 2013), where the authors show that CDH5 is specifically activated in CD133+ GSC and it is essential for transdifferentiation of GSC into endothelial cells. ANGPT1 also has been shown to be expressed by glioma cells and as a regulator of endothelial spreading and angiogenesis (Audero E et al., 2000). However, we agree that we had not closed the loop, causally linking the altered expression of these genes and radiation-induced vascular conversion. To address whether CDH5 and ANGPT1 play a causative role in radiation-induced vascular conversion, we performed knockdown of both CDH5 and ANGPT1 in gliomaspheres *in vitro* and show that the vascular-like conversion is reduced in the absence of these lineage markers in response to radiation (Figure 5J and K.) and added these new results to the manuscript as follows:

“Next, we examined whether CDH5 and ANGPT1 played a functional role in radiation-induced vascular conversion. Knockdown of either CDH5 or ANGPT1 with lentiviral expression of shRNAs reduced the expression of key endothelial and pericyte markers induced by radiation, respectively. However, CDH5 knockdown did not alter pericyte marker expression, and ANGPT1 knockdown did not significantly alter endothelial markers, suggesting that these genes act as lineage markers of distinct vascular-cell states, and promote radiation-induced endothelial- and pericyte-like transdifferentiation. “

4. The authors own data led them to examine trophic effects of these vascular-like glioma cells. Yet pursuit of the function of these trophic factors *in vivo* is not well-developed. For example, the authors do not mention whether P300 blockade has any impact on the trophic factors that are apparently responsible for tumor regrowth after radiation or whether depletion of sub-populations of glioma cells also reduces the presence of the trophic factors.

Response: We agree with the reviewer that proving the functional effects of the trophic factors in tumor progression would be significant and clinically relevant. However, to demonstrate the functional effects of each trophic factor secreted by induced GEC and GPC *in vitro* as well as

vivo, and determining the mechanism by which they do so and validating them would take months to years and would be well beyond the scope of this paper. We will pursue this in our future investigation. We have addressed the latter point and present evidence that blocking P300 activity with C646 not only inhibits vascular conversion, but also reduces expression of these trophic factors as shown by RNA-sequencing and validation by qRT-PCR of the factors that had growth-promoting effect on radiated cells. We also show that expression of iGEC and iGPC-derived factors is diminished post-radiation in P300-KD cells. (Fig 6J, Fig S6P and Fig 7F, shown below).

5. The apparent disconnect between the function of the vascular-lineage genes in glioma cells, the trophic factors, chromatin accessibility, and glioma rebound after radiation is a confusing aspect of these work that makes the overall theme/message of the manuscript hard to appreciate.

Response: We have modified the text to clearly establish a link between these steps. We show that radiation-mediated vascular-like conversion requires P300-induced increase in chromatin accessibility of CDH5 and ANGPT1 lineage markers. Knockdown of P300 or CDH5 or ANGPT1 reduces vascular-like conversion and expression of trophic factors that promote tumor recurrence post-treatment.

6. A major issue with the paper is lack of clarity or information or consistency on the experimental aspects of the work (number of repeats, number of mice, etc). In some cases, the statistical test chosen appears to be incorrect or needs clarification (see below). This has made it challenging to judge the merits of the manuscript overall and it is not clear how robust the data are in total.

Response: We appreciate the reviewer for pointing out these errors in our description of statistics which we have now corrected and included the relevant numbers of replicates, mice etc for each figure in their respective figure legends and in the methods.

Fig 1 It is not clear how many samples were analyzed (how many repeats?). Is this just one sample being assayed? The hallmark angiogenesis genes are going down over time? Also, to this reviewer, there is a lot of extraneous information in this figure that make it hard to follow overall; examples include numerous GO-terms with no apparent connection to the study at large (or at least this has not been well-explained to the reader).

Response: For the *in vitro* single cell sequencing experiment, we pooled the cells for sequencing from 3 replicates each from radiated and non-radiated gliomaspheres from a primary patient-derived GBM line, and have mentioned the number of cells analyzed in the text

as well as in the figure. For the *in vivo* single cell sequencing experiment in Figure S1, we pooled FACS sorted GFP+ tumor cells from 5 radiated and 5 non-radiated tumors and analyzed by sequencing. We included several GO-terms even though they are not directly relevant to the point being made because we wanted to show that radiation-induced gene expression changes is not limited to vascular conversion, but could also contribute to other processes affecting phenotype plasticity.

Fig 2 The images in panels H and K could be improved with confocal images of the individual channels. Functional parameters of the blood vessels in question should also be assessed. Again (and this is a theme in almost every figure), information on the number of experimental repeats (technical versus biological) are lacking in most panels. Just one example is panel D. All data should be at least n=3 independent biological replicates and individual data points should be shown. Panels I and J have only n=2. Can appropriate statistics be done on n=2?

Response: We have replaced the images in Panels H and K with better quality images as well panels I and K. For Panel D, we performed the experiments with at least 3 biological replicates, and 3 independent experiments (N=3). For panels I and J, N was stated as 2, because it was performed as two independent experiments but with 5 mice per experiment. We have now rectified this in the figure legend in the revised manuscript.

Fig 3 Ac-LDL images are not convincing. It is not clear what panel F is showing. Again, how many repeats were done for the experiments? How many mice per group? How many sections were analyzed? How were these data quantified in an unbiased way? It is hard to assess whether any of these data are rigorous or not. The Matrigel tube forming assay is not really a robust indicator of vascular-like potential. It is not clear whether a patent lumen is present. Many cells, even fibroblasts, will self-organize into transient “tube-like” structures when plated at certain densities on Matrigel or other ECM, but this is no indication that fibroblasts can form bona fide blood vessels.

Response: We agree that the matrigel tube formation assay alone is not an indicator of EC-like behavior, and that is why we have added multiple assays such as LDL-uptake and matrigel plug to show their EC-like potential. Panel F shows vessels formed post implantation of plugs under the skin, with a few iGEC incorporated into vessels. We have included the statistics on number of mice, replicates per condition in the figure legend in the revised manuscript.

Fig 4 Data in B and D appear modest. For example, the increase in trophic factors is quite minimal. Only RNA is assayed. How about protein levels? Do the levels of these factors reach relevant physiologic levels/activities *in vivo*? Again, how many repeats were done here? Three biological replicates in duplicate or triplicate should be done. Do the glioma cells express the receptors for these trophic factors? In its present form, links between an increase in these trophic factors and glioma survival are correlative. It is hard to follow the data in panel F and G. Again, how many repeats? Is this n=1 experiment with n=5 mice? This would not appear to be robust enough to assess whether the results are rigorous.

Response: We agree that the data in B and D are modest, and this is because induced GEC and GPC secrete several growth promoting factors, and we do not expect any single factor to increase growth robustly. The increase in mRNA expression of trophic factors in induced GEC and GPC is quite high with fold changes (Log Fc ranging from 5-10) more than in radiated tumor cells. Heatmap shows the Row Z-score of expression of each gene between groups. The

number of replicates and statistics is included in the figure legend in the revised manuscript. We have not assayed for protein levels for these factors or tested for their activity in vivo, which will be considered in our future work.

Fig 6 Panel B is n=2? How can appropriate statistics be carried out? Why was a t-test used in C? These data should be assessed using an ANOVA since there are multiple variables/comparisons. The same is true for other graphs in this figure. Individual data points should be shown as well (mean value +/- STD for minimum of 3 biological replicates ideally assayed in duplicate or triplicate).

Response: This is an error on our part in labelling appropriate N for each figure. This is now corrected and indicated in the legend. Panel B (western quantification was performed with N=3). For Fig 6C, we did perform ANOVA followed by post hoc t-test, and this has been rectified now in the revised manuscript.

Fig 7 Are the trophic factors identified in this study also impacted by the p300 inhibitor? How many mice in G? How many repeats?

Response: We have now performed a quantitative RT-PCR post P300 knockdown as well as C646 inhibition to show that blocking P300 also reduces the trophic factors made by iGEC and iGPC (Fig 6J, Fig S6P and Fig 7F). We have indicated the number of mice per group (N=5 mice per condition) in the figure legend in the revised manuscript.

Fig S1 How have the authors distinguished bona fide endothelial cells from glioma-derived endothelial-like cells or pericytes in the scRNA-seq analysis? How was the purity of these samples assayed? Panel J is not convincing, nor is it quantified. Some of this staining almost looks non-specific to this reviewer w/o adequate negative controls or high quality images showing membrane staining of Cdh5 or Pecam.

Response: For *in vitro* scRNA-seq, the gliomaspheres were cultured for several passages in GBM media that only allows selective propagation of GSC and tumor cells. Other cell types including endothelial cells and pericytes are selected against and do not propagate in the GBM media. Thus, the endothelial-like and pericyte-like cells we observed in RNA-sequencing can only be derived from the GBM cells. This was further validated by our experiments in which we depleted reporter-positive cells prior to radiation. For *in vivo* scRNA-seq of xenografts, we FACS sorted the GFP+ tumor cells from the brains of mice and hence excluded contamination from other cell types including the vascular cells. Purity and viability of the cells was assayed by FACS sorting for GFP+ (viable cells) and DAPI (dead cell exclusion) in the *in vivo* experiment, and cell viability by Trypan blue staining for *in vitro* experiment. We have included higher quality images to show membrane staining of VE-CADHERIN/CDH5 and other markers in in the revised manuscript, and as shown below (Fig S1J).

Fig S2 How many biological replicates? Individual data points should be shown for all graphical data. In panel N there is no legend callout. What is this showing?

Response: We have included the information on N in the figure legend. In the previous version of the manuscript, Panel N was immunostaining of CD31 and α SMA with GFP to label vascular cells and tumor cells in the murine GBM model. This panel is now labelled as panel P in the revised manuscript, and a figure legend has been added. All qPCR and flow cytometry experiments were performed with 3 technical replicates in 3 independent experiments. For CD133 *in vivo* experiment, we used N=3 mice per group. For murine GBM model *in vivo* experiment, we used N=5 mice per group.

Fig S3 Ac-LDL images are not convincing. DiI-Ac-LDL uptake by endothelial cells typically has a punctate appearance by microscopy.

Response: We have repeated the experiment and included higher magnification images showing the punctate appearance of LDL in Fig 3 and Fig S3.

Fig S4 How many mice? How many biological replicates.

Response: All mice experiments were performed with at least N=5 mice per group, and we have included the information on N in the figure legend of the revised manuscript.

Fig S5 In panel B, none of these pathways appear to support vascular-like features of glioma cells – is that what is meant to be shown here? Panel D is not convincing. The results are very modest and based on the images, it is not clear how these data were quantified in a rigorous or unbiased way. Were ROIs chosen? Were multiple tumors imaged? How were these data normalized to account for differences in tumor size?

Response: Figure S5 panel B shows gene sets associated with mesenchymal development and transition as pericytes are mesenchymal cells and are derived from a neural crest lineage. We also show stem cell-related gene sets to make the point that radiation-induces phenotype conversion of tumor cells to stem-like cells and vascular-like cells. For Figure S5D, Quantitation of the staining in the tumor images is explained in detailed in the methods section. Briefly, cell counts were measured randomly in 5 different regions per tumor and 5 tumors in each condition (Radiated and non-radiated) in a blinded fashion.

Fig S6 For some panels, how many biological/technical replicates?

Response: For qRT-PCR and flow cytometry experiments measuring gene expression and reporter expression, N=3 replicates per condition, and 3 independent experiments. We have included the information on N in the figure legend in the revised manuscript.

Reviewer #3 (Remarks to the Author):

Muthukrishnan et al. treated gliomaspheres and glioma xenografts with ionizing radiation (IR) and profiled specimens from those experiments with single-cell and bulk transcriptomics and immunofluorescence. These studies were complemented with in vivo angiogenesis, in vivo treatment with histone acetyltransferase inhibitor assays, and several other in vitro functional assays. The authors claim that IR causes a shift to a mesenchymal phenotype in vitro and in vivo, that IR induces endothelial- and pericyte-like phenotypes in glioma stem cells. They claim that depleting cells with a vascular tumor phenotype enhances survival, that mechanistically vascular conversion is regulated by histone acetylation of vascular genes and that targeting histone acetyltransferase reduces vascular conversion.

The study is timely and addresses several current topic in glioma biology. The approach is thoughtful, the experiments are generally well controlled, the bioinformatics analysis is rigorously done. In many ways, the manuscript is largely complete.

There are several minor to moderate weaknesses that diminish enthusiasm slightly. The authors may want to address the following points:

1. The reporting on the glioma-spheres and xenografts should be improved. This ties into the role of potential epistatic effects and generalizability of the authors conclusions that appear to be based on a single glioma culture. There is considerable inter-patient heterogeneity in GBM and it appears that the authors have used cultures from a single patient sample for all figures except figure 3. How many distinct patient samples were used to derive the cells for scRNA-

seq/RNA-seq/IF? What is the genotype of these cultures, do they have EGFRviii, NF1, P53 or other mutations that would affect how we interpret mesenchymal signatures or IR responses? Validating some of the basic correlations in additional cultures with different genetics and phenotypes would strengthen the manuscript.

Response: Single-cell analysis, bulk RNA-sequencing and immunostaining was performed on a single patient-derived gliomasphere line (Proneural subtype, EGFR amplified) in Figure 1 and 2. We then validated the vascular-lineage gene induction by radiation in a total of 12 patient-derived GBM lines spanning both Proneural and Mesenchymal subtype (in vitro, GBM gliomasphere lines are often restricted to two subtypes; Laks, et al., 2016) as well primary and recurrent GBM with different mutations--including EGFRVIII pos (HK308, HK412) , EGFRVIII neg (HK217, HK382, HK372, HK347) EGFR amplification (HK408, HK412, HK372), PTEN loss (HK382, HK417, HK217, HK347), monosomy 10 (HK408, HK412, HK413, HK417) and polysomy 7 (HK413))and also IDH-mutant tumors (HK213, HK252)-- by QRT-PCR and reporter activation, as well as LDL uptake and matrigel tube formation. This is shown in Figure 2 and Figure S2.

2. The authors show that ionizing radiation induces mesenchymal transition and vascular conversion. Prior studies implicated TNF-alpha in mesenchymal transition following IR, with NFKB activation being essential. Are these pathways, and in general prior studies of IR-induced mesenchymal transition, supported by the authors' data? Do the authors have any evidence for the mechanism of IR induced vascular conversion?

Response: In this study, we checked for mesenchymal markers predominantly because pericytes are derived from a mesenchymal lineage, and we do see a small increase in the canonical mesenchymal markers previously reported in the literature. The mechanism we report here is that radiation induces an increase in chromatin accessibility in CDH5 (for endothelial-conversion) and ANGPT1 (for pericyte conversion) through P300. We have added new data to show that knockdown of P300 or CDH5 or ANGPT1 reduces the vascular-like conversion. Also, with regard to NFKB and TNFA, we do not see an upregulation in the gliomasphere line we studied (shown below).

3. In the authors' data, is vascular conversion strictly tied to mesenchymal transition? Do most of the vascular cells in their glioma-spheres and xenografts express markers of mesenchymal cells?

Response: While we do tie the pericyte conversion to mesenchymal transition as mesenchymal GBM lines express higher pericyte-like markers, we think that EC- like conversion is distinct from mesenchymal transition. The xenografts were performed using a proneural line, and the pericyte-like cells do express mesenchymal markers, however EC-like cells do not. Our RNA-sequencing and qRT-PCR results show that induced GEC do not express mesenchymal markers or gene sets, but induced GPC do, further confirming that they are distinct cell-lineages.

4. Evidence suggests that the mesenchymal phenotype is inducible and that mesenchymal cells can convert to other phenotypes. So, the mesenchymal state may represent a transient state, is this the case for vascular-like states? Blocking P300 HAT inhibits vascular conversion in the first place, but can the vascular phenotype be regressed?

Response: We have considered whether vascular-like state is transient. However, we are able to isolate and sort these cells and expand them in culture without losing their marker expression indicating that it may not be a transient phenotypic state. We have checked for marker expression 14d, 21d and 30d post radiation, and they still express the markers (data not shown in the paper). There are also other studies (Hu et al., 2016; De Pascalis et al., 2018) using paired GBM patient samples that show that GBM cells expressing endothelial markers increase in recurrent (post-treatment) tumors which implies that it is not a transient phenotype. We do not yet know if C646 treatment post-radiation regresses the vascular phenotype.

5. When only one shRNA is used per target, as in Figure 7, it is natural to worry about off-target effects.

Response: We only presented data for the ShRNA EP300_A that had the most robust effect in vitro in the paper. We did test 4 different constructs of shRNA targeting P300, and only two showed significant downregulation of P300 mRNA. We have included the data for shEP300_D in the supplementary Fig S7A, D, E as shown below.

6. The survival analysis in Figure 4G is somewhat surprising, in that 1) co-injection with vascular cells and no radiation does better than unsorted cells. 2) co-injection with vascular cells and no radiation does the same as vascular-depleted (mC-) cells without radiation. Do the authors have a comment on this?

Response: Our in vitro experiments with CM from non-induced cells also did not have a significant growth effect on tumor cells (Fig 4), only the induced cells did. We observed the same effect in vivo. This implies that trophic factors made by induced GEC and GPC have a significant growth effect only on radiated tumor cells, and not on non-radiated tumors.

7. In Figure 4H-J the authors report tumor growth after 6 weeks, but not overall survival?

Response: Unfortunately, during this experiment, some of our tumor-bearing animals died suddenly, before they became ill, which did not allow for sufficient numbers of animals to be analyzed for survival. We feel that the main outcome of the experiment, tumor growth, revealed the most important findings. Furthermore, our findings with diphtheria toxin receptor-bearing

animals yielded essentially identical results—that elimination of the radiation-converted vascular cells inhibited tumor growth in radiation-treated animals. We show that co-transplantation of transdifferentiated cells with radiated tumor cells increases tumor burden and reduces survival (Fig 4G), so we would expect that their depletion would improve survival.

8. There is a strong preclinical rationale for HDAC inhibition in GBM. This is essentially the opposite of blocking HAT or at least antagonistic. I would ask the authors to address this in the discussion and I am curious to hear their comment on this.

Response: We agree that there is rationale for HDAC inhibition in select GBM patients, although clinical trials have been somewhat disappointing. However, we do not believe that HDAC inhibition has been tested in the peri-radiation period. The reviewer is correct in that we might expect to see the opposite effect of HDAC inhibition on radiation-induced vascular specification, but our preliminary experiments with broad-spectrum HDAC inhibitors did not verify this. The reasons underlying this are likely highly complex and would warrant a completely separate future investigation.

Reviewer #4 (Remarks to the Author):

Glioblastoma is a challenging disease to treat, in part because standard radiation and chemotherapy are insufficient to eradicate the tumor cells. In fact recurrence after radiation therapy, is not well understood. The authors who jest that radiation stress and this is a dynamic shift in glioma cells to acquire vascular and parasite like cell phenotypes. They suggest that this allows for trophic support and proliferation of tumor cells the selective depletion of which reduces tumor growth. They suggest that blocking P300 histone acetyltransferase activity reverses these epigenetic changes induced by radiation. Therapeutic targeting of P300 the author suggest will inhibit the therapy induced adaptive response.

Strengths of the project include the focus on GSC that can convert under certain stress conditions (eg radiation) and may ultimately be responsible for recurrence, in particular the endothelial and pericyte transition. The breadth of information across single cell in bulk transcriptome over many patient-derived glioma sphere cultures, orthotopic xenografting, further support the strength of their results. Overall the thoughtful composition of experiments with detailed controls and biostatistical analysis highlight the significance of these results.

1. The main limitation includes reflection on primary human samples or correlation with patient survival or grade in response to radiation

Response: Our study was based on the prior observation that GEC-like cells were found in significantly higher numbers in recurrent tumors compared to matched primary tumors (Hu et al., 2016 and De pascalis et al., 2018) and they contribute to tumor recurrence. Even the GPC-like conversion was found to be significantly high in patient tumors and targeting these cells inhibited tumor growth post-chemotherapy (Zhou et al., 2017). Obtaining tissue immediately or in the weeks following radiation is a daunting challenge, as tumors are not generally biopsied at this time and is beyond the scope of the current manuscript.

2. The contribution of radiation-induced phenotypic plasticity and GBM was mainly performed using single cell RNA sequencing of primary glioma sphere cell lines. How do these specific data then reflect on expression or ATACseq analysis on recurrent (radiation resistant) glioblastoma?

Response: This is a good question. We examined both primary and recurrent gliomaspheres and, in this small sample size (N=6, primary and N=4 Recurrent), did not see a difference in radiation-induced vascular conversion. In fact, we wouldn't necessarily expect there to be such a difference, as they are grown as gliomaspheres prior to testing and one might predict that we are actually excluding the radiation-induced vascular cells from these cultures. More telling would be a direct comparison of primary and recurrent samples from the same patient, a study that we hope to carry out, even despite the caveats that recurrence is often long after radiation. We have yet to perform ATAC-sequencing on recurrent GBM tumor tissue.

3. The initial data both in the tumor spheres as well as in NSG mice lack the contribution of the immune component. Is there any suggestion either in data from this group or otherwise that this could confound your results? Likewise, patient derived glioma spheres represent a subset of selected cells that can be maintained or proliferate under these conditions. The incorporation of a single syngeneic mouse model validation experiment is helpful but these concerns remain.

Response: We agree that studies with more mouse models could contribute, but we hope that the reviewer will agree that the work included is sufficient to prove the point that the presence of an intact immune system does not block the process of radiation-induced vascular conversion. Future studies will investigate the interaction of this process with immunotherapy.

4. Given that P300 disruption could dysregulate other mechanisms of DNA repair after radiation, the precise mechanism in glioma remains unknown. C646, a selective small molecule inhibitor of P300 is a known radiosensitizer in other cancers.

Response: We acknowledge that P300 could affect other mechanisms of DNA repair, but our knockdown and C646 inhibitor experiments show robust effects predominantly on vascular gene induction indicating that it has a significant role in this process.

REVIEWERS' COMMENTS

Reviewer #2 (Remarks to the Author):

The Authors have answered to the suggestions and criticisms raised by the Reviewer.

Reviewer #3 (Remarks to the Author):

The authors have done a nice job addressing the comments and I have no further queries.

Reviewer #4 (Remarks to the Author):

The authors have addressed all of my concerns.